# Place Cells as Multi-Scale Position Embeddings: Random Walk Transition Kernels for Path Planning

**Minglu Zhao**[1][⋆], **Dehong Xu**[1][⋆], **Deqian Kong**[1][⋆], **Wen-Hao Zhang**[2],**Ying Nian Wu**[1]

[1]UCLA    [2]UT Southwestern Medical Center

## Abstract

The hippocampus supports spatial navigation by encoding cognitive maps through collective place cell activity. We model the place cell population as non-negative spatial embeddings derived from the spectral decomposition of multi-step random walk transition kernels. In this framework, inner product or equivalently Euclidean distance between embeddings encode similarity between locations in terms of their transition probability across multiple scales, forming a cognitive map of adjacency. The combination of non-negativity and inner-product structure naturally induces sparsity, providing a principled explanation for the localized firing fields of place cells without imposing explicit constraints. The temporal parameter that defines the diffusion scale also determines field size, aligning with the hippocampal dorsoventral hierarchy. Our approach constructs global representations efficiently through recursive composition of local transitions, enabling smooth, trap-free navigation and preplay-like trajectory generation. Moreover, theta phase arises intrinsically as the angular relation between embeddings, linking spatial and temporal coding within a single representational geometry.

## 1 Introduction

Place cells in the hippocampus are central to spatial cognition, firing selectively at specific locations as animals navigate their environment [1, 2]. Rather than modeling individual place fields, we propose viewing place cell populations as non-negative position embeddings derived from spectral decomposition of multi-step transition kernels that collectively encode spatial relationships. This population-level approach captures the hippocampus's role in forming cognitive maps, integrating metric and topological properties for flexible navigation [3].

Hippocampal place cells exhibit dynamic behaviors, adapting firing fields to environmental changes [4], scaling along the dorsoventral axis [5], and engaging in preplay before exploration [6]. These properties suggest a multi-scale, adaptive representation that supports navigation across diverse contexts—from narrow passages to open terrains. The hippocampus's ability to predict novel paths, as in preplay, and maintain robust navigation under environmental deformations further underscores its computational sophistication [7–9].

We formalize these insights by modeling place cell population as vector embeddings derived from a multi-step symmetric random walk. The transition matrix admits spectral decomposition, enabling construction of non-negative embeddings $h(x, \tau)$ such that the inner product between embeddings approximates the normalized transition probability: $\langle h(x, \tau), h(y, \tau) \rangle = q(y|x, \tau)$, where $h(x, \tau)$ is the embedding at location $x$, and $q(y|x, \tau)$ is the symmetric transition probability over time step $\tau$. Remarkably, combining this inner product structure with non-negativity and orthogonality constraints

---

⋆Equal Contribution.

Project page: https://sites.google.com/view/place-cells.

Code: https://github.com/mingluzhao/Place_Cell.

induces emergent sparsity—the embeddings naturally develop disjoint support sets, explaining why place cells exhibit localized firing fields without explicit regularization. The time parameter $\sqrt{\tau}$ defines a spatial scale hierarchy, mirroring hippocampal dorsoventral scaling [5]. $q(y|x, \tau)$ defines spatial adjacency between $x$ and $y$ at scale or resolution $\sqrt{\tau}$, and the pairwise adjacency relationships $(q(y|x, \tau), \forall x, y)$ are reduced into individual embeddings $(h(x, \tau), \forall x)$ that collectively form a multi-scale, Euclideanized, and sparsified cognitive map of the environment. The Euclideanization emerges from inner product expansion of the transition kernel, the sparsification arises naturally from non-negativity of the place cell responses and the orthogonality between embeddings with zero inner product, and the multi-scale hierarchy enables adaptive navigation across spatial resolutions. Efficient matrix squaring ($P_{2\tau} = P_\tau^2$) computes global transitions from local ones ($P_1$) without past trajectory memorization, enabling preplay-like shortcut detection [10, 7].

Our framework uses gradient ascent on $q(y|x, \tau) = \langle h(x, \tau), h(y, \tau) \rangle$ with adaptive scale selection, choosing the time scale maximizing the gradient for trap-free, smooth trajectories. This produces robust navigation with properties like boundary avoidance, diffraction-like passage guidance, aligning with hippocampal navigation [11, 9, 3]. Continuous interpolation ensures smooth gradient fields, supporting natural paths. Due to the Euclideanization, path planning amounts to gradient descent on Euclidean distance at the selected scale, enabling what can be called "straight forward" path planning.

Beyond spatial coding, we propose a novel theoretical framework where theta phase naturally emerges from the population embedding structure, unifying spatial representation with temporal dynamics and offering a principled account of theta phase precession.

Appendix A provides an extensive review of related work.

Our contributions include: (1) reconceptualizing place cells as non-negative embeddings from spectral decomposition encoding transition probabilities at multiple scales; (2) demonstrating emergent sparsity explaining localized place fields; (3) modeling multi-scale, Euclideanized, and sparsified cognitive maps via time scale parameter $\sqrt{\tau}$ for transition probabilities; (4) employing efficient matrix squaring to build up multi-scale spatial relationships; (5) introducing adaptive gradient ascent for Euclideanized path planning at selected scale; (6) proposing theta phase formulation from population embeddings; and (7) demonstrating biological plausibility through properties like preplay. Bridging connectionist models [12, 13] and cognitive map theories [2, 9], our framework offers a scalable, biologically inspired model of hippocampal spatial navigation.

## 2 Method

### 2.1 Multi-Step Random Walk Transition Kernel

The foundation of our approach is a symmetric random walk on a discrete lattice over the environment (e.g., a $40 \times 40$ lattice), with a subset of lattice points belonging to the obstacles. This random walk serves as a mapping policy rather than a goal-reaching policy. Remarkably, this purely random exploration policy leads to optimal path planning without explicit policy optimization. We use $\tau$ to denote the time step of this random walk mapping policy, to avoid confusion with the time $t$ of the planned trajectory. $\tau$ plays the role of scale, which is adaptively selected during path planning.

For a location $x = (i, j)$ on a 2D lattice, let $N(x)$ be the set of its unobstructed neighbors, i.e., neighbors that do not belong to obstacles (e.g., 4 nearest neighbors), we define the one-step transition probabilities as:

- For each unobstructed neighbor $y$ of $x$: $p(y|x, \tau = 1) = p_{\text{move}}$ (e.g., 1/4 in the case of 4 nearest neighbors)
- For self-transition: $p(x|x, \tau = 1) = 1 - |N(x)| \cdot p_{\text{move}}$

where $|N(x)|$ is the number of unobstructed neighbors of $x$. A critical property of this formulation is that the transition probabilities are symmetric, meaning $p(y|x, \tau = 1) = p(x|y, \tau = 1)$ for all locations $x$ and $y$. This symmetry ensures that the random walk process preserves the bidirectional nature of spatial relationships, which is essential for creating a well-defined proximity metric.

The above random walk defines $\tau$-step symmetric transition kernel $p(y|x, \tau)$ over time step $\tau$, which measures the spatial adjacency between $x$ and $y$ at a spatial scale or resolution captured by $\tau$.

## 2.2 Heat Diffusion, Geodesic Distance and Topological Connectivity

Our discrete random walk model establishes a connection to the continuous heat equation. For small spatial discretization $dx$ and temporal discretization $dt$ with $dx = \sqrt{dt}$, our discrete random walk converges to the heat diffusion equation with reflecting boundary conditions as $dx \to 0$ [14, 15]:

$$\frac{\partial p(y|x,\tau)}{\partial \tau} = \alpha \nabla^2 p(y|x,\tau) \quad \text{on} \quad \Omega \setminus \Omega_{\text{obstacles}} \tag{1}$$

$$\frac{\partial p(y|x,\tau)}{\partial n} = 0 \quad \text{on} \quad \partial\Omega_{\text{obstacles}} \tag{2}$$

where $\Omega$ is the whole region, $\Omega_{\text{obstacles}}$ is the region of obstacles, and $\partial\Omega_{\text{obstacles}}$ is its boundary. $\alpha$ is the diffusion coefficient (different one-step transition $p(y|x,\tau = 1)$ in discrete case leads to different $\alpha$, e.g., $\alpha = 1/4$ for 4-nearest neighbor random walk).

A fundamental result from heat diffusion theory relates the short-time behavior of the heat kernel to geodesic distance. For small values of $\tau$, the heat kernel has the asymptotic form [16, 17]:

$$p(y|x,\tau) \approx \frac{1}{4\pi\alpha\tau} \exp\left(-\frac{d_g^2(x,y)}{4\alpha\tau}\right) \tag{3}$$

where $d_g(x,y)$ is the geodesic distance. For open domains, $d_g(x,y)$ becomes Euclidean distance, and $p(y|x,\tau) \sim \mathcal{N}(x, 2\alpha\tau)$, a Gaussian distribution with variance $2\alpha\tau$ or standard deviation $\sqrt{2\alpha\tau}$, demonstrating that $\sqrt{\tau}$ correspond to the spatial scale. We take square root of $\tau$ to emphasize this scaling relationship between time and space.

This result demonstrates that $d_t(x,y) = -\tau \log q(y|x,\tau)$ approximates the squared geodesic distance for small values of $\tau$, providing a fundamental connection between random walks and geodesic distances in complex environments.

As $\tau$ increases, the transition probability incorporates additional information about path multiplicity and global connectivity, creating a multi-scale representation of spatial relationships. In particular, for large $\tau$, the transition probability is dominated by eigenvalues that are close to 1, whose eigenvectors depend on topological connectedness rather than local geometry.

Appendix D provides detailed background on heat equation. Appendix G explains topological properties for large $\tau$.

## 2.3 Place Cells as Non-negative Spectral Embeddings

We formalize place cell population as vector embedding $h(x,\tau) \in \mathbb{R}^n$, where $n$ is the number of place cells (e.g., $n = 500$). The inner product between embeddings approximates the normalized transition probability kernel:

$$\langle h(x,\tau), h(y,\tau) \rangle \approx q(y|x,\tau) \tag{4}$$

where $q(y|x,\tau) = p(y|x,\tau)/\sqrt{p(x|x,\tau) \cdot p(y|y,\tau)}$ is the normalized transition probability, so that $\|h(x,\tau)\| = 1$ due to normalization. For each cell $i$, $h_i(x,\tau) \geq 0$ for biological plausibility, and $h_i(x,\tau)$ is the response map or profile of cell $i$ at spatial scale $\sqrt{\tau}$.

The above formulation can be derived through spectral decomposition of the transition matrix. Since the one-step transition matrix $P_1$ is symmetric by construction, its powers $P_\tau = P_1^\tau$ admit an eigendecomposition [18]: $P_\tau = Q\Lambda^\tau Q^T$, where $Q$ is orthogonal ($Q^T Q = I$) and $\Lambda = \text{diag}(\lambda_1, \ldots, \lambda_n)$ contains eigenvalues $0 \leq \lambda_i \leq 1$ (assuming the random walk is irreducible and aperiodic).

From this spectral decomposition, we can construct a spectral embedding by defining: $H_i(x,\tau) = \lambda_i^{\tau/2} Q_i(x)$ where $Q_i$ is the $i$-th column of $Q$. This yields an exact representation of the transition probability through inner products:

$$p(y|x,\tau) = \sum_i H_i(x,\tau) H_i(y,\tau) = \langle H(x,\tau), H(y,\tau) \rangle \tag{5}$$

where $H_i(x,\tau)$ is the $i$-th element of the vector $H(x,\tau)$. For the normalized transition probability $q(y|x,\tau)$, we define normalized embeddings:

$$h_{\text{spec}}(x,\tau) = \frac{H(x,\tau)}{\|H(x,\tau)\|} = \frac{H(x,\tau)}{\sqrt{p(x|x,\tau)}} \tag{6}$$

which satisfy:

$$\langle h_{\text{spec}}(x,\tau), h_{\text{spec}}(y,\tau) \rangle = \frac{p(y|x,\tau)}{\sqrt{p(x|x,\tau) \cdot p(y|y,\tau)}} = q(y|x,\tau) \tag{7}$$

$p(x|x,\tau)$ is constant in the open field and is smooth in general, so $q$ is essentially a scaled version of $p$.

However, these spectral embeddings may contain negative components, conflicting with the biological constraint that neural firing rates must be non-negative. Horn's theorem [19, 20], which is built upon the above spectral decomposition, guarantees the existence of a non-negative $h(x,\tau)$, such that $\langle h(x,\tau), h(y,\tau) \rangle = q(y|x,\tau)$ for non-negative matrix factorization.

This formulation represents a fundamental shift from modeling individual place cells to modeling the place cell population as distributed position embedding. The embedding vector $h(x,\tau)$ represents the firing rates of all place cells at location $x$ for spatial scale $\sqrt{\tau}$, capturing the idea that it is the pattern across the population—not the activity of any single cell—that encodes location.

The time parameter $\sqrt{\tau}$ serves as a fundamental unit of spatial resolution or scale, with larger values producing broader, more diffuse representations, and smaller values producing more localized representations. This naturally mirrors the variation in place field sizes observed along the dorsoventral axis of the hippocampus [5, 21].

See Appendix C for more details on spectral analysis. Appendix E provides analytical results for open field, where $q(y|x,\tau)$ is Gaussian, and elements of $h(x,\tau)$ exhibit Gaussian profiles over $x$.

## 2.4 Matrix Squaring, Learning, and Continuation

Let $P_1$ be the one-step transition matrix for the random walk on a discrete lattice. $P_1$ depends on obstacles in the environment and amounts to local one-step exploration, as detailed in subsection 2.1.

We calculate $P_\tau$ for a discrete set of $\tau$, $\mathcal{T} = \{\tau = 2^k, k = 1, ..., K\}$, via $P_{2\tau} = P_\tau^2$. The matrix squaring is very efficient for calculating $P_\tau$ for $\tau \in \mathcal{T}$, and these $P_\tau$ correspond to explorations of different spatial scales $\sqrt{\tau}$, where the adjacent spatial scales in $\mathcal{T}$ have a ratio $\sqrt{2}$, mirroring grid cell module scaling ($\sim$1.4–1.7) [22]. In addition to matrix squaring, one can design any discrete sequence $\mathcal{T} = \{t_k, k = 1, \ldots, K\}$, and calculate $P_\tau$ for $\tau \in \mathcal{T}$ using matrix multiplication.

We learn a separate population of place cells $h(x,\tau)$ for each $\tau \in \mathcal{T}$, by minimizing the least squares error:

$$\mathcal{L} = \sum_{x,y} [q(y|x,\tau) - \langle h(x,\tau), h(y,\tau) \rangle]^2 \tag{8}$$

We learn $h(x,\tau)$ over the discrete lattice, where $\sum_{x,y}$ in (8) is over all pairs of points on the lattice. We optimize this objective using the AdamW optimizer [23], where after each iteration, for each $x$, we set the negative elements of $h(x,\tau)$ to 0, and then normalize $h(x,\tau)$ so that $\|h(x,\tau)\| = 1$.

The learning reduces pairwise adjacency relationships ($q(y|x,\tau), \forall x, y$) into individual embeddings ($h(x,\tau), \forall x$), which collectively form a map of the environment.

After learning, we can use bi-linear interpolation to make $h(x,\tau)$ a continuous map over $x$. As a result, $q(y|x,\tau) = \langle h(x,\tau), h(y,\tau) \rangle$ also becomes continuous, approximating the transition kernel of the continuous heat equation. The continuous $q(y|x,\tau)$ can then elegantly guide path planning in continuous space instead of discrete lattice.

## 2.5 Euclideanized Cognitive Map

Geometrically, $\langle h(x,\tau), h(y,\tau) \rangle$ is the cosine of the angle between the normalized vectors $h(x,\tau)$ and $h(y,\tau)$. Moreover, we have

$$\frac{1}{2}\|h(x,\tau) - h(y,\tau)\|^2 = 1 - \langle h(x,\tau), h(y,\tau) \rangle = 1 - q(y|x,\tau) \tag{9}$$

That is, the angle or the Euclidean distance between $h(x,\tau)$ and $h(y,\tau)$ encodes proximity or adjacency between $x$ and $y$. Thus, our method geometrizes transition probability.

$(h(x, \tau), \forall x)$ is a 2D manifold in the high-dimensional embedding space. This 2D manifold is an embedding of the 2D physical space. Because of the flexibility endowed by the high-dimensional embedding space, the path between $x$ and $y$ on the manifold is essentially "Euclideanized" even though the physical path is far from being Euclidean.

## 2.6 Emergent Sparsity from Non-negativity and Orthogonality

The conjunction of non-negativity and orthogonality induces emergent sparsity in the representational code. Let $h(x, \tau) \in \mathbb{R}_+^n$ denote the representation at position $x$ and scale $\sqrt{\tau}$, such that $\langle h(x, \tau), h(y, \tau) \rangle = q(y|x, \tau)$, where $q(y|x, \tau)$ measures spatial adjacency at scale $\sqrt{\tau}$. If two positions $x$ and $y$ are non-adjacent ($q(y|x, \tau) = 0$), the corresponding vectors must be orthogonal. Because all components of $h(x, \tau)$ are non-negative, orthogonality implies disjoint support—no component can be active in both representations. Consequently, the joint requirement of non-negativity and orthogonality forces most components to be zero, yielding sparse, localized activity patterns. As a result, each place cell fires at a localized place, and the population of place cells collectively tile the environment. The symbolic representation thus emerges automatically from population-based representation.

The scale parameter $\tau$ modulates this sparsity: for small $\tau$, most position pairs are orthogonal, producing highly localized "place fields"; for large $\tau$, overlaps increase and place fields broaden. Thus, the locality of place-cell activation follows directly from the geometry of non-negative, inner-product spectral embeddings. Appendix F provides technical details.

## 2.7 Straight Forward Path Planning with Adaptive Scale Selection

We use adaptive gradient following for goal-directed navigation. When navigating from a current position $x$ to a target location $y$, we select the next position $x_{\text{next}}$ from the neighborhood $\partial(x)$ where $\partial(x) = \{z : z = x + \Delta r (\cos \theta, \sin \theta)\}$, $\Delta r$ is the step size, and $\theta$ is discretized into equally spaced direction in $[0, 2\pi)$. We use $\partial(x)$ for continuous $x$ for path planning to differentiate from $N(x)$ in the discrete lattice for random walk.

The path planning algorithm is as follows:

- Compute the gradient of the normalized transition probability for each neighbor $z \in \partial(x)$ and each scale $\tau$:

$$\Delta(z, \tau) = q(y|z, \tau) - q(y|x, \tau) = \langle h(y, \tau), h(z, \tau) \rangle - \langle h(y, \tau), h(x, \tau) \rangle \quad (10)$$

- Select the scale $\tau^*$ that provides the strongest directional signal:

$$\tau^* = \arg\max_{\tau \in \mathcal{T}} \max_{z \in \partial(x)} \Delta(z, \tau) \quad (11)$$

- Choose the next position that maximizes the gradient at the selected scale:

$$x_{\text{next}} = \arg\max_{z \in \partial(x)} \Delta(z, \tau^*) \quad (12)$$

Note that in the above algorithm, $x$ and $z$ are continuous, because $h(x, \tau)$ is made continuous in $x$ with bi-linear interpolation after learning on discrete lattice.

$\Delta(z, \tau)$ measures the reduction in the angle or squared Euclidean distance. The path planning seeks maximal reduction in angle or Euclidean distance. Therefore we call it the straight forward path planning in the Euclideanized embedding space, where the vector $h(x, \tau)$ rotates straightly to $h(y, \tau)$ on the path.

This approach selects the scale $\tau^*$ that provides the clearest guidance for the current navigation step. Intuitively, larger scales provide better guidance for distant goals, while smaller scales offer more precise navigation for nearby goals. The adaptive scale selection mechanism automatically finds this optimal scale at each step, similar to choosing the most appropriate "ruler" for measuring at the current distance.

Now consider the idealized continuous limit where $\Delta r \to 0$, $\theta$ is continuous, and $\tau$ is continuous. The path planning algorithm follows the gradient $\nabla_x q(y|x, \tau) = \nabla_x q(x|y, \tau)$ where

$q(y|x, \tau) = q(x|y, \tau)$ due to symmetry. The planed trajectory is the gradient ascent flow: $dx(t)/dt = \nabla_x q(x(t)|y, \tau(t))$, where $t$ is the time on the planed trajectory, $\tau(t) = \arg\max_\tau \|\nabla_x q(x(t)|y, \tau)\|$ is the optimal scale at time $t$ with the maximal gradient.

The gradient-based navigation framework has the following key properties that ensure computational efficiency and biological plausibility:

1. Adaptive scale selection dynamically adjusts $\sqrt{\tau^*} \propto d(x, y)$ for precise navigation in the open environment, mirroring hippocampal spatial tuning [11].

2. The unimodal smooth gradient field $\nabla_x q(x|y, \tau)$, with a unique maximum at the goal, ensures trap-free paths, aligning with hippocampal navigation [11, 9].

3. Planned paths match the shortest path for small $\tau$, but prioritize topological connectivity for large $\tau$, reflecting cognitive map robustness [9, 3].

4. Near obstacles, $\nabla_x q(x|y, \tau)$ flows parallel to boundaries, preventing collisions, akin to hippocampal obstacle avoidance [11].

5. Diffraction-like patterns guide trajectories through passages, resembling hippocampal maze navigation [11, 9].

6. Topological invariance maintains navigation under environmental deformations, mirroring hippocampal place cell encoding [3, 9].

7. Matrix squaring from local transitions predicts shortcuts, embodying hippocampal preplay [7, 8].

These properties, detailed in Appendix G, underpin the framework's alignment with neural mechanisms.

## 2.8 Theta-Phase Procession Based on Angle-Phase Duality

The hippocampus not only encodes spatial relationships but also organizes temporal dynamics through theta phase precession, where place cells fire at specific phases of the theta rhythm as an animal traverses their fields [24, 25]. We extend our population embedding framework to model this phenomenon.

As explained above, the non-negativity and inner product structure of the position embeddings induces localized sparse patterns, i.e., each place cell only fires at a localized field around a place, and together the place fields of all the place cells tile the environment. Let $\mu_i = \arg\max_x h_i(x, \tau)$, i.e., the center of the place field for cell $i$. The theta phase of cell $i$ can be defined in terms of the angle between $h(x, \tau)$ and $h(\mu_i, \tau)$. As $x$ approaches $\mu_i$, the angle changes from $\pi/2$ to 0, and as $x$ moves away from $\mu_i$, the angle changes from 0 to $\pi/2$. Appendix H provides details.

## 2.9 Integrating Grid Cells

For scientific reductionism, we focus on place cells without incorporating grid cells. A natural extension would explore the relationship $h(x, \tau) = W(\tau)g(x)$, where $g(x)$ represents the vector of grid cell activations and $W(\tau)$ is a learned transformation matrix, potentially unifying our framework with the complementary roles of place and grid cells in spatial representation [9]. Appendix I provides the formulation.

# 3 Experiments

We design experiments to evaluate both the biological plausibility of our position embedding framework and its functional capabilities in path planning. All experiments are conducted in a simulated environment. We first examine the positional representation and population density profiles in an open field. Then we create more complex environments by adding obstacles. For implementation details, see Appendix J; For additional experiment visualizations, see Appendix K.

## 3.1 Place Cell Representations in Open Field

We begin our numerical experiments in a simple open field environment to demonstrate fundamental properties. The environment consists of a $40 \times 40$ lattice grid. For the transition kernel, we

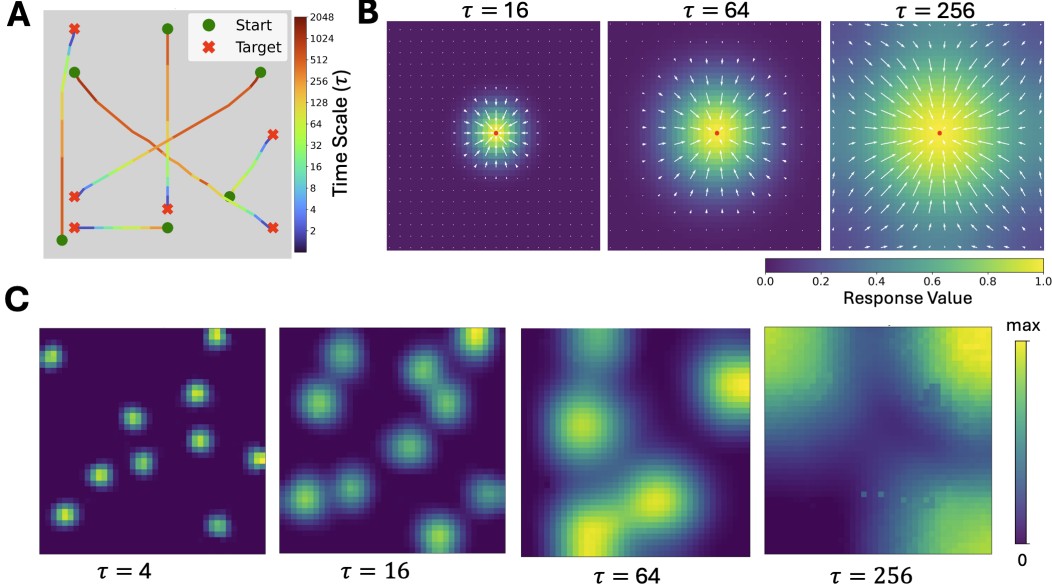

Figure 1: Place Cell Representations and Navigation in Open Field Environment. (A) Goal-directed path planning trajectories with adaptive scale selection (selected scale is color coded, red for big scale and blue for small scale). (B) Normalized transition probability kernels $q(y|x,\tau)$ at multiple scales with gradient vector fields. $y$ is fixed at the center of the environment. (C) Learned activation patterns of $h(x,\tau)$ at different scales across randomly chosen cells, exhibiting Gaussian-like firing fields centered at specific locations within the open environment.

employ a $3 \times 3$ neighborhood, meaning each position has at most 8 neighboring points, with transition probability $p_{\text{move}} = 1/9$ distributed uniformly among neighbors as well as self. Within this environment, we learn $h(x,\tau)$ embeddings across all lattice points. For hyperparameters, the total number of place cells is set to 500. The temporal dynamics are captured across multiple scales using the time parameter $\tau = 2^k$, where $k = 1, 2, ..., 11$, allowing us to analyze system behavior across spatial scales.

### 3.1.1 Multi-Scale Transitions

To evaluate the normalized transition probability kernel $q(y|x,\tau)$, we visualize the learned Gaussian-like patterns across multiple temporal scales in Figure 1 (B). Each panel depicts the transition probability distribution for time scales $\tau = 16, 64, 256$ with position $y$ fixed at the environment center.

The gradient fields reveal a critical temporal dependency: as $\tau$ increases, gradient magnitudes intensify for locations distant from the center, while maintaining directional integrity toward the target. This multi-scale property is particularly significant for path planning applications, as it enables a multi-resolution navigational framework.

### 3.1.2 Spatial Activation Patterns

Our model is trained by minimizing Equation 8. The optimization results demonstrate remarkable fidelity in approximating normalized transition probabilities through inner products of population embedding vectors with correlation coefficients above 0.9 for all scales. Figure 1 (C) illustrates the learned activation patterns of $h(x,\tau)$ at scales $\tau = 4, 16, 64, 256$ for randomly chosen cells. The spatial receptive fields of individual units emerge naturally from our training process, exhibiting the characteristic Gaussian-like firing fields centered at specific locations within the open environment. This spatial tuning closely resembles the well-documented properties of hippocampal place cells observed in rodent navigation studies [1, 2].

### 3.1.3 Path Planning and Adaptive Scale Selection

To evaluate the navigational capabilities of our model, we implemented the gradient-based path planning. We randomly selected start and target locations within the open field environment. At each step, the agent evaluates potential next positions by sampling orientation $\theta$ from 36 equally spaced directions within $[0, 2\pi)$ and a fixed radius $\Delta r$ of one grid unit. Even though our model is learned in a $40 \times 40$ discrete lattice, in path planning, we can reach continuous positions $x$, where the representation of its location $h(x, \tau)$ is calculated by linear interpolation of the 4 closest neighbors. Movement direction is determined by maximizing the gradient $q(y|z, \tau) - q(y|x, \tau)$ where $y$ is the target location, $x$ is the current position, and $z$ represents each candidate next position. And for each step, we choose the $z$ with the largest gradient. Our results in Figure 1 (A) demonstrate that the learned model consistently generates near-optimal trajectories.

A key feature of our position embedding model is the adaptive scale selection mechanism, where $\tau^*$ is selected with maximal gradient. The mechanism is illustrated in Figure 1 (A), where different colors represent the selected time scale $\tau^*$ at each navigational step. This reveals a systematic progression from coarse to fine scales as the agent approaches its goal. This pattern emerges naturally from the gradient fields associated with different scales.

When navigating to distant goals, larger scales ($\tau \geq 128$) are selected, which provides clearer guidance for long-range planning. As the agent reaches medium distances from the goal, the selected scales transition to intermediate values ($\tau = 16$ to $\tau = 64$), balancing directional guidance with increasing spatial resolution. In the final approach phase, the system converges on the smallest available scales ($\tau = 2$ to $\tau = 8$), which offered the most precise local guidance.

This adaptive scale selection mirrors the progressive engagement of different regions along the dorsoventral axis of the hippocampus during navigation, as observed in rodent studies [5, 21].

## 3.2 Place Cells in Complex Environments

To investigate the robustness and adaptability of our method, we extend our analysis to complex environmental geometries that more closely resemble naturalistic navigation scenarios. These environments incorporate obstacles and boundaries that fundamentally alter the adjacency relationships between locations, requiring the model to learn representations that respect environmental constraints rather than simple Euclidean distances.

### 3.2.1 Environment and Experiment Setup

To evaluate performance across different environments, we implement three distinct mazes. Detailed visualization is shown in Figure 2.

- U-shaped Maze: A corridor structure with a single 180-degree turn, creating a simple non-convex navigation challenge.

- S-shaped Maze: A serpentine corridor with multiple turns, requiring longer detour paths around obstacles.

- Four-room Maze: A compartmentalized environment with four chambers connected by narrow doorways, representing a hierarchically structured space.

To accommodate environmental complexity, we modify our random walk dynamics to incorporate spatial constraints imposed by obstacles and boundaries. For any location $x$, transitions to neighboring locations $y$ that are prohibited by the obstacles setting $p(y|x, \tau = 1) = 0$ for these forbidden transitions. Consequently, the self-transition probability is adjusted to maintain normalization $p(x|x, \tau = 1) = 1 - |N(x)| \cdot p_{\text{move}}$. The key difference between open field and obstacle-containing environments lies in the value of $N(x)$. In open fields, $N(x)$ consistently includes all 8 neighboring locations (except at boundaries). With obstacles, $N(x)$ represents only the subset of accessible neighboring locations from position $x$, which could be smaller than 8. We maintain $p_{\text{move}} = 1/9$ across all environments for consistency. Despite these modifications to the underlying transition dynamics, our learning approach remains consistent. We compute the normalized transition probabilities $q(y|x, \tau)$ based on these constraint-respecting dynamics and train our model to learn the embedding function $h(x, \tau)$ using the same objective function as in the open field.

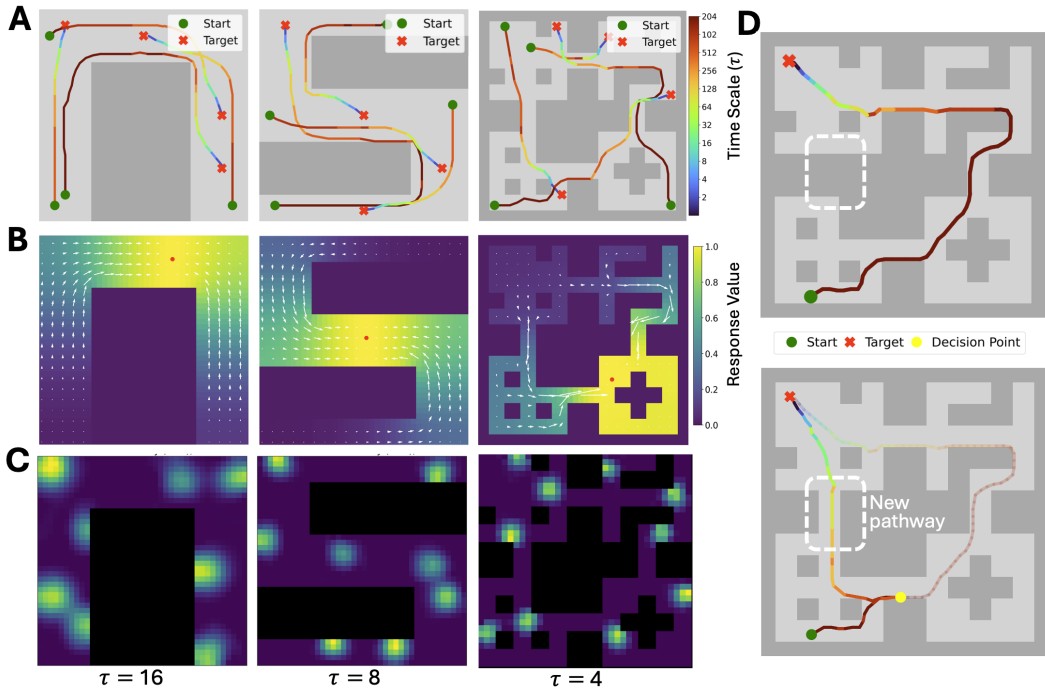

Figure 2: Place Cells in Complex Maze Environments. (A) Path planning through obstacle-containing environments. (B) Topologically-informed transition kernels $q(y|x, \tau)$ with gradient fields. Target $y$ is marked as a red point. (C) Randomly sampled place cell profiles at multiple spatial scales. (D) Remapping with environmental modification.

### 3.2.2 Navigation Efficiency

We evaluated our model's navigation capabilities across multiple environments, conducting 50 trials per environment with randomly sampled start and goal positions from non-obstacle areas. Our model achieved 100% success rate across all tested environments.

We compare against the Bug algorithm [26] as a baseline since it prioritizes finding feasible paths rather than strictly optimizing for the shortest path. The Bug algorithm moves directly toward the goal until encountering an obstacle, then follows the obstacle's boundary until it can resume its direct path. We implement the oracle-enhanced version [27] that knows which boundary-following direction yields the shortest path for each obstacle, establishing an upper bound on performance.

To quantify efficiency, we use Success weighted by inverse Path Length (SPL) [28], where values near 1.0 indicate near-optimal paths and values exceeding 1.0 suggest our method found shortcuts the Bug algorithm missed. Details of SPL calculation appear in Appendix J. In simpler environments (U-shape), our method performs comparably to the oracle-enhanced Bug algorithm. However, in complex environments (S-shape and Four-room), our approach significantly outperforms the baseline, discovering more efficient paths that the Bug algorithm fails to identify even with oracle assistance.

Table 1: Path planning results.

| Environment | Success | SPL ($\uparrow$) |
|---|---|---|
| Open field | 100% | $0.991 \pm 0.04$ |
| U-shape | 100% | $0.919 \pm 0.25$ |
| S-shape | 100% | $1.392 \pm 0.98$ |
| Four-room | 100% | $1.519 \pm 1.17$ |

Table 2: Path planning in Four-Room environment.

| Metrics | Bug (w/ Oracle) | Bug (w/o Oracle) | A* Search | Random Walk | Our Method |
|---|---|---|---|---|---|
| Success (%) | 100% | 14% | 100% | 2% | 100% |
| SPL ($\uparrow$) | $0.66 \pm 0.51$ | $0.15 \pm 0.00$ | $1.08 \pm 0.50$ | $0.01 \pm 0.00$ | $1.00$ |

We also compared our method with several different baselines in the Four-Room environment, with each trial limited to 50,000 steps. The results in Table 2 show that baseline approaches like the Bug algorithm (without oracle guidance) and Random Walk fail to solve most of the challenging scenarios.

### 3.2.3 Transition and Learned Profile in Complex Environments

To evaluate the adaptive capabilities of our approach, we examine the transition kernel $q(y|x, \tau)$ and the resulting place cell activation profiles $h(x, \tau)$ in environments containing complex obstacles. Figure 2 (B) presents the normalized transition probability distributions $q(y|x, \tau)$ for strategically selected locations, with a large scale to reveal long-range spatial relationships. The diffraction-like patterns, where probability flow encounters obstacles, reveal how spatial information propagates through available pathways while respecting environmental constraints.

The gradient fields illustrate navigation potential, showing how an agent could efficiently pass obstacles. Importantly, these transition probabilities capture the true topological structure of complex environments rather than simple Euclidean distances, a critical property for realistic navigation where direct paths are often blocked. The emergent probability gradients provide a natural mechanism for guiding optimal path planning that automatically adapts to environmental geometry.

We visualized place cell profiles at multiple spatial scales across complex maze environments in Figure 2 (C). Each panel displays a small subset of non-overlapping place cells at different scales $\tau$, emphasizing the emergent sparsity and localized nature of individual receptive fields. Despite increased environmental complexity, the learned place cell population maintains comprehensive spatial coverage throughout accessible regions. At smaller scales, place fields are tightly localized; at larger scales, fields broaden while maintaining their localized tiling structure. This multi-scale coverage property demonstrates the model's robust ability to develop effective spatial representations regardless of environmental geometry, a critical feature for reliable navigation in diverse settings.

### 3.2.4 Remapping Properties

We evaluated our model's ability to adapt to environmental modifications, analogous to the remapping phenomenon observed in rodent place cells. Previous neurobiological studies have shown that hippocampal place fields reorganize in geometry-dependent ways when familiar environments are altered through elongation or restructuring.

We conducted a two-phase experiment with the environment shown in the upper panel of Figure 2 (D). After training our model to convergence (2000 iterations) on this initial configuration, we modified the arena by removing several obstacles to introduce a novel shortcut path, as shown in the lower panel of Figure 2 (D). The intervention created a minimal physical alteration but significantly changed the environment's topological structure.

We then fine-tuned our pre-trained model on this modified environment with just 50 iterations at a reduced learning rate (5e-4), allowing the position embeddings to adapt efficiently to the new spatial configuration. When testing path planning to identical target locations, the fine-tuned model successfully identified and utilized the newly available shortcuts at critical decision points where pathways diverged.

## 4 Conclusion

This paper reconceptualizes hippocampal place cells as population embeddings approximating multi-step random walk transition kernels, where $\langle h(x, \tau), h(y, \tau) \rangle = q(y|x, \tau)$, offering a biologically plausible model of spatial navigation [29, 30]. The time parameter $\sqrt{\tau}$ defines a multi-scale representation, mirroring dorsoventral place field scaling [5], adaptability [4], and remapping [31]. Gradient ascent on $q(y|x, \tau)$ with adaptive scale selection produces trap-free trajectories, guided by boundary avoidance, diffraction-like passage navigation, aligning with hippocampal cognitive maps [11, 9, 3]. Efficient matrix squaring ($P_{2\tau} = P_\tau^2$) computes global transitions from local ones ($P_1$) without past trajectory memorization, enabling preplay-like shortcut detection [10, 7, 8]. Bridging connectionist models [12, 13] and cognitive map theories [2, 9], our framework captures hippocampal navigation's dynamic properties, providing a scalable, computationally efficient model for complex environments.

## Acknowledgments

Y. W. is partially supported by NSF DMS-2415226, DARPA W912CG25CA007 and research gift funds from Amazon and Qualcomm.

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

# Contents

Sections and subsections that contain mostly background materials are so marked in the titles. The remaining sections and subsections contain novel developments.

## A  Related Work

### A.1  Place Cell Models and Representations

The study of hippocampal place cells has a rich history since their discovery by [1]. Computational models of place cells have evolved from simple Gaussian tuning curves [29] to more sophisticated approaches. Several models have explored population-level representations of place cells, including manifold embeddings [32] and latent space models [33]. However, these approaches typically treat place cells as separate entities encoding specific locations rather than as collective embeddings encoding transition probabilities.

Matrix factorization approaches to neural population activity have been applied to various brain regions [34, 35], though rarely with the specific mathematical connection to random walk processes proposed in our work. Recent work by [13] proposed a successor representation framework for place cells, which shares some conceptual similarities with our transition probability approach but differs in the specific mathematical formulation and implementation.

Hidden Markov Models (HMMs) have also been applied to model hippocampal spatial coding. The Clone-Structured Causal Graph model (CSCG) [36] treats space as a latent sequence and uses a structured graph with cloned nodes to disambiguate aliased sensory observations in different contexts. Related work [37] further develops sequence-based models of hippocampal function.

### A.2  Multi-scale Spatial Representations

The variation in place field sizes along the dorsoventral axis of the hippocampus has been extensively documented [5, 21, 22], but computational models that explicitly address this multi-scale organiza-

tion remain limited. Models incorporating scale in place cell representations include hierarchical approaches [38], wavelet-like representations [39], and scale-space theories [40] borrowed from computer vision.

Our approach differs by deriving the multi-scale representation directly from the time parameter of a random walk process, providing a principled connection between scale and exploration time that has not been previously exploited in place cell models.

### A.3 Navigation and Path Planning

Biologically-inspired navigation algorithms have drawn on various hippocampal properties. Vector-based navigation models [41, 42] and successor representation approaches [38, 43, 44] have demonstrated effective navigation capabilities. Diffusion-based path planning algorithms [45, 46] share mathematical similarities with our heat equation formulation but lack the direct connection to neural representations.

Recent work by [47] and [48] has emphasized the role of population coding in navigation but without the specific inner product relationship and transition probability framework we propose.

### A.4 Successor Representation

Our work is closely related to the Successor Representation (SR) [49, 50], as both frameworks build upon powers of a transition matrix to construct a representation of space. The SR is typically defined as a discounted sum, $M = \sum_{k=0}^{\infty}(\gamma T)^k$, which yields the expected discounted future occupancy of all states from any given starting state, and the discount factor $\gamma$ implicitly sets a single, predictive temporal horizon. In contrast, our approach utilizes the discrete powers of the transition matrix directly, $P_\tau = P^\tau$, to explicitly model the transition probabilities at multiple, distinct time horizons, where the parameter $\tau$ corresponds directly to the time scale of a random walk. This multi-scale representation aligns more closely with the observed functional gradient of place field sizes along the hippocampal dorsoventral axis.

The primary novelty of our framework lies in learning a biologically-constrained, non-negative matrix factorization of these transition kernels, where vector embeddings $h(x, \tau)$ are learned such that their inner product reconstructs the transition probability, $\langle h(x, \tau), h(y, \tau) \rangle = q(y|x, \tau)$. A crucial theoretical insight is that the combination of this inner product objective with non-negativity and orthogonality constraints for distant locations gives rise to **emergent sparsity**; the embeddings are forced to have disjoint support sets, providing a geometric explanation for the localized firing fields of place cells without explicit regularization. This multi-scale architecture confers significant functional advantages, enabling an adaptive gradient-ascent-based navigation policy that selects the optimal scale $\tau^*$ at each step to generate smooth, trap-free trajectories. This approach is not only more scalable, but also offers a more direct navigation mechanism that bypasses the need for explicit value function computation common in SR-based reinforcement learning agents.

### A.5 Inner Product Spaces in Neural Representation

Inner product spaces as a basis for neural computation have been explored by several researchers [51, 52]. More recently, [53] proposed that neural populations represent probability distributions through their activation patterns, with some conceptual overlap with our approach.

Navigational planning in inner product spaces has connections to kernel methods in machine learning [54] and information geometry [55], though these connections have been underexplored in neuroscience. Our work bridges these fields by explicitly relating inner products between place cell populations to transition probabilities derived from random walk processes.

### A.6 Heat Equation and Diffusion Models in Neuroscience

The connection between neural dynamics and diffusion processes has been explored in various contexts [56]. The specific relationship between heat kernels and geodesic distances, which forms a foundation for our approach, has strong connections to manifold learning [57] and dimensionality reduction techniques [58].

Our work builds upon these ideas by applying them specifically to place cell populations and spatial navigation, providing a novel bridge between diffusion processes, neural representations, and navigational behavior.

## B  Limitations

### B.1  Lack of Sensory Integration

Our current model does not incorporate sensory inputs for spatial navigation[11, 2]. Real hippocampal place cells integrate multimodal sensory information including visual landmarks, self-motion cues from path integration, and boundary detection. This sensory integration is fundamental to how place cells form their spatial receptive fields and anchor representations to environmental features.

Without sensory processing, our model cannot explain how place fields emerge during initial exposure to novel environments. In our framework, position embeddings $h(x, \tau)$ are learned from pre-defined transition probabilities that already encode environmental structure, rather than being built up through sensory experience. We also cannot model perceptual anchoring, where place cells maintain stable firing relative to visual landmarks, or how sensory conflicts (such as navigation in darkness) affect place cell stability. Future extensions incorporating vision-based inputs, self-motion integration, and boundary vector cell signals would significantly enhance biological plausibility.

### B.2  Limited to 2D Static Environments

Our experiments and analyses are restricted to two-dimensional static environments, whereas biological spatial navigation operates in three-dimensional space and continuously changing contexts[5, 59]. Studies in flying bats have revealed volumetric place cells encoding 3D positions, and terrestrial animals navigating multi-level structures require three-dimensional representations. Extension to 3D introduces complications our framework does not address: vertical movements and gravity-dependent asymmetries, different place field scaling across dimensions, and substantially increased computational complexity. Dynamic environments would require time-varying transition probabilities and continuous updating mechanisms our framework does not provide.

### B.3  Online Sequential Learning

Another limitation is our lack of online learning over sequential experiences to model temporal dynamics and memory consolidation[7, 8]. Our batch optimization learns $h(x, \tau)$ across all spatial locations simultaneously, fundamentally differing from how biological place cells develop through experience. Biological place cell formation is incremental: initial weak or unstable spatial tuning gradually sharpens through repeated exploration via activity-dependent synaptic plasticity. Our framework does not capture this gradual emergence from unstructured initial conditions. Our matrix squaring $P_{2\tau} = P_\tau^2$ achieves global integration instantaneously based on environmental structure, not through iterative learning over sleep-wake cycles analogous to biological replay.

Despite these limitations, our framework offers valuable theoretical insights into computational principles underlying hippocampal spatial navigation, particularly the role of multi-scale transition probabilities, emergence of sparsity from geometric constraints, and the connection between random walk dynamics and cognitive maps.

## C  Spectral Analysis of Multi-Step Random Walk

This appendix provides a detailed spectral analysis of the multi-step random walk transition kernel, establishing connections between our random walk model and position embeddings.

### C.1  Eigendecomposition of the Transition Matrix (Background)

Since the one-step transition matrix $P_1$ is symmetric by construction, it admits an eigendecomposition:

$$P_1 = Q\Lambda Q^T, \tag{13}$$

with orthogonal $Q$ ($Q^T Q = I$) and diagonal $\Lambda = \text{diag}(\lambda_1, \ldots, \lambda_n)$, where $0 \leq \lambda_i \leq 1$, where we assume the random walk is irreducible and aperiodic. The multi-step transition matrix is:

$$P_1^\tau = Q\Lambda^\tau Q^T \tag{14}$$

The eigenvalues are typically ordered as $1 = \lambda_1 > \lambda_2 \geq \lambda_3 \geq \ldots$, with the first eigenvalue corresponding to the stationary distribution and subsequent eigenvalues capturing spatial patterns at increasing levels of detail.

## C.2  Position Embeddings from Spectral Decomposition

The spectral decomposition provides a natural position embedding. If we define:

$$H_i(x, \tau) = \lambda_i^{\tau/2} Q_i(x) \tag{15}$$

in the discrete case where $Q_i(x)$ is the $i$-th column of $Q$, or:

$$H_i(x, \tau) = e^{\lambda_i \tau/2} \phi_i(x) \tag{16}$$

in the continuous case (where $\lambda_i$ and $\phi_i$ are respectively eigenvalues and eigenfunctions of the Laplacian), then the transition probability can be expressed as:

$$p(y|x, \tau) = \sum_i H_i(x, \tau) H_i(y, \tau) = \langle H(x, \tau), H(y, \tau) \rangle \tag{17}$$

This provides a closed-form expression for position embeddings that exactly reproduce the transition probabilities through inner products.

## C.3  Normalization and Non-Negative Embeddings

To obtain normalized embeddings, we define:

$$h_{\text{spec}}(x, \tau) = \frac{H(x, \tau)}{\|H(x, \tau)\|} = \frac{H(x, \tau)}{\sqrt{p(x|x, \tau)}} \tag{18}$$

This normalized embedding satisfies:

$$\langle h_{\text{spec}}(x, \tau), h_{\text{spec}}(y, \tau) \rangle = \frac{p(y|x, \tau)}{\sqrt{p(x|x, \tau) \cdot p(y|y, \tau)}} = q(y|x, \tau) \tag{19}$$

However, $h_{\text{spec}}(x, \tau)$ may contain negative components, which conflicts with the biological constraint that neural firing rates must be non-negative. This is where Horn's theorem becomes relevant.

## C.4  Non-Negative Matrix Factorization (Background)

Horn's theorem [19] provides the theoretical foundation for obtaining non-negative embeddings from our transition matrices: If $A$ is a symmetric matrix with non-negative entries, then there exists a non-negative matrix $B$ such that $A = BB^T$.

We provide a sketch of the proof:

*Proof sketch of Horn's theorem.* Given a symmetric non-negative matrix $A$, consider its spectral decomposition $A = U\Sigma U^T$. Let $A = \sum_i \sigma_i u_i u_i^T$, where $\sigma_i$ are eigenvalues and $u_i$ are eigenvectors.

For each outer product $u_i u_i^T$ (which may have negative entries), we can express it as a linear combination of non-negative rank-1 matrices: $u_i u_i^T = \sum_j c_j v_j v_j^T$, where $v_j$ are non-negative vectors and $c_j$ are coefficients.

By appropriate selection of $v_j$ (e.g., using vertices of the hypercube defined by the signs of $u_i$), we can ensure that $c_j \geq 0$ when $\sigma_i > 0$. This allows us to express $A$ as a sum of non-negative rank-1 matrices, which can be arranged as $A = BB^T$ where $B$ has non-negative entries. $\square$

For our normalized transition matrix $Q_\tau = [q(y|x, \tau)]$, which is symmetric with non-negative entries, Horn's theorem guarantees the existence of a non-negative matrix $H_\tau$ such that $Q_\tau = H_\tau H_\tau^T$. The rows of $H_\tau$ provide our desired non-negative position embeddings $h(x, \tau)$.

# D  Heat Diffusion with Reflecting Boundaries (Background)

This appendix provides a detailed mathematical derivation of the connection between our discrete random walk model and the continuous heat equation, establishing the relationship between transition probabilities and geodesic distances.

## D.1  From Discrete Random Walk to Reflecting Heat Equation

We begin with a discrete random walk on a two-dimensional integer grid. For simplicity, we assume the one-step transition $p(y|x, \tau = 1)$ of the random walk is to move to one of 4 nearest neighbors with $p_{\text{move}} = 1/4$. We assume this simplest $p(y|x, \tau = 1)$ in our theoretical derivations in all the relevant sections in the Appendix. Similar results can be obtained for more general $p(y|x, \tau = 1)$, with a different diffusion coefficient $\alpha$.

Let $(i, j) \in \mathbb{Z}^2$ denote discrete spatial coordinates, and $k \in \mathbb{Z}_{\geq 0}$ denote discrete time steps. To connect with continuous diffusion, we introduce spatial and temporal units:

$$x = i \cdot dx, \quad y = j \cdot dx, \quad \tau = k \cdot dt \tag{20}$$

where $dx$ is the spacing between adjacent grid points and $dt$ is the time step. Following standard diffusion scaling, we set $dx = \sqrt{dt}$, which ensures convergence to a well-defined limit as $dx \to 0$ [14].

The discrete random walk has the following transition probabilities:

$$p((i', j')|(i, j), k = 1) = p_{\text{move}} = \frac{1}{4} \quad \text{for each unobstructed neighbor } (i', j') \text{ of } (i, j) \tag{21}$$

$$p((i, j)|(i, j), k = 1) = 1 - N(i, j) \cdot p_{\text{move}} = 1 - \frac{N(i, j)}{4} \tag{22}$$

where $N(i, j)$ is the number of unobstructed neighbors of location $(i, j)$ (maximum 4 in a 2D grid with 4-connectivity).

## D.2  Derivation of the Heat Equation

To derive the continuous limit, for interior points (those not adjacent to obstacles), the discrete update rule is:

$$\begin{aligned}
p(i, j, k+1) &= p(i, j, k)(1 - 4p_{\text{move}}) + p_{\text{move}}[p(i+1, j, k) + p(i-1, j, k) + p(i, j+1, k) + p(i, j-1, k)] \\
&= p(i, j, k) + p_{\text{move}}[p(i+1, j, k) + p(i-1, j, k) + p(i, j+1, k) + p(i, j-1, k) - 4p(i, j, k)]
\end{aligned} \tag{23}$$

Dividing both sides by $dt$ and using $p_{\text{move}} = 1/4$:

$$\frac{p(i, j, k+1) - p(i, j, k)}{dt} = \frac{1}{4} \cdot \frac{1}{dt}[p(i+1, j, k) + p(i-1, j, k) + p(i, j+1, k) + p(i, j-1, k) - 4p(i, j, k)] \tag{24}$$

Using the standard finite difference approximation for the Laplacian [60]:

$$\nabla^2 p(i, j, k) \approx \frac{1}{dx^2}[p(i+1, j, k) + p(i-1, j, k) + p(i, j+1, k) + p(i, j-1, k) - 4p(i, j, k)] \tag{25}$$

Substituting $dx^2 = dt$ and taking the limit as $dt \to 0$:

$$\lim_{dt \to 0} \frac{p(i, j, k+1) - p(i, j, k)}{dt} = \frac{1}{4}\nabla^2 p(x, y, t) \tag{26}$$

This yields the heat equation with diffusion coefficient $\alpha = 1/4$:

$$\frac{\partial p(x, y, \tau)}{\partial \tau} = \alpha \nabla^2 p(x, y, \tau) \tag{27}$$

## D.3 Reflecting Boundary Conditions

Our discrete random walk enforces reflecting boundary conditions, preventing probability flow into obstacles, mirroring hippocampal obstacle avoidance [11]. Consider a boundary point $(i, j)$ with an obstacle at $(i+1, j)$, so the number of valid neighbors is $N(i, j) = 3$ (points $(i-1, j)$, $(i, j+1)$, $(i, j-1)$). The self-transition probability is $p_{\text{stay}} = 1 - \frac{N(i,j)}{4} = \frac{1}{4}$, redistributing probability that would flow to the obstacle back to $(i, j)$.

The probability update for $(i, j)$ at time step $k + 1$ is:

$$p(i, j, k + 1) = \frac{1}{4} p(i, j, k) + \frac{1}{4} \left[ p(i-1, j, k) + p(i, j+1, k) + p(i, j-1, k) \right] \tag{28}$$

This can be rewritten as:

$$p(i, j, k + 1) = p(i, j, k) + \frac{1}{4} \left[ p(i-1, j, k) + p(i, j+1, k) + p(i, j-1, k) - 3p(i, j, k) \right] \tag{29}$$

In finite difference methods, the reflecting condition $\left. \frac{\partial p}{\partial n} \right|_{(i+1,j)} = 0$ is enforced using a ghost point at $(i+1, j)$, setting $p(i+1, j, k) = p(i, j, k)$ to ensure zero normal flux [60]. Substituting into the interior Laplacian update (equation 30):

$$p(i, j, k+1) = p(i, j, k) + \frac{1}{4} \left[ p(i-1, j, k) + p(i+1, j, k) + p(i, j+1, k) + p(i, j-1, k) - 4p(i, j, k) \right] \tag{30}$$

yields equation (29), as $p(i+1, j, k) = p(i, j, k)$ reduces the neighbor terms to three. This substitution ensures the boundary update aligns with the random walk's mechanism, where $p_{\text{stay}} = \frac{1}{4}$ assigns zero probability to obstacle transitions, maintaining probability conservation and enabling smooth navigation around obstacles, as observed in hippocampal place cell activity [11].

## D.4 Connection to Geodesic Distance

Varadhan's formula [16] establishes a deep relationship between the heat kernel and geodesic distance. For a complete Riemannian manifold $M$ with heat kernel $p(x, y, \tau)$, Varadhan proved that:

$$\lim_{\tau \to 0} -4\tau \log p(x, y, \tau) = d_g^2(x, y) \tag{31}$$

where $d_g(x, y)$ is the geodesic distance between points $x$ and $y$.

While Varadhan's original large deviation principle [16] applies to smooth manifolds without boundaries, extensions to domains with reflecting boundaries [61] ensure that the short-time behavior of the heat kernel $p(y, \tau | x, 0)$ reflects the geodesic distance $d_g(x, y)$, the shortest path within $\Omega$ avoiding obstacles. For our normalized transition probability: a similar asymptotic relationship holds:

$$\lim_{\tau \to 0} -\tau \log q(y | x, \tau) = \frac{d_g^2(x, y)}{4\alpha}, \tag{32}$$

where $\alpha = 1/4$ is the diffusion coefficient. This follows because $p(x | x, \tau)$ and $p(y | y, \tau)$, influenced by boundary reflections, have asymptotic forms that cancel in the logarithm as $\tau \to 0$, leaving the geodesic term dominant.

As $\tau$ increases, our distance metric transitions from approximating geodesic distance to incorporating more global aspects of the domain's connectivity, creating a multi-scale representation that seamlessly integrates local metric information with global connectivity structure. Section G explains eigen analysis of connectivity.

# E  Open-Field Environment

In unbounded, obstacle-free environments (open fields), the symmetric random walk simplifies to an isotropic diffusion process, offering a theoretical justification for the Gaussian tuning of place cells

observed in such settings [1, 29]. As the grid discretization refines ($dx \to 0$, $dt \to 0$, $dx = \sqrt{dt}$), the transition probability $p(y|x, \tau)$ converges to the heat equation's fundamental solution in 2D free space:

$$p(y|x, \tau) = \frac{1}{4\pi\alpha\tau} \exp\left(-\frac{\|y - x\|^2}{4\alpha\tau}\right) \tag{33}$$

where $\alpha = 1/4$ is the diffusion coefficient, and $\|y - x\|^2$ is the squared Euclidean distance. Since $p(x|x, \tau) = \frac{1}{4\pi\alpha\tau}$ is position-independent, the normalized transition probability becomes:

$$q(y|x, \tau) = \frac{p(y|x, \tau)}{\sqrt{p(x|x, \tau)p(y|y, \tau)}} = \exp\left(-\frac{\|y - x\|^2}{4\alpha\tau}\right) = \exp\left(-\frac{\|y - x\|^2}{\tau}\right) \tag{34}$$

This Gaussian kernel, with variance $\sigma^2 = 2\alpha\tau = \tau/2$, reflects the diffusive spread of the random walk and mirrors the approximately Gaussian firing fields of hippocampal place cells in open environments. Given $q(y|x, \tau) = \langle h(x, \tau), h(y, \tau)\rangle$, the position embeddings $h(x, \tau)$ must reproduce this Gaussian decay. We construct $h(x, \tau)$ with Gaussian components, providing a mathematical basis for why place cell population activity exhibits Gaussian profiles in open fields.

**Theorem 1** (Gaussian Embeddings in Open Fields). *In an unbounded 2D open field, where* $q(y|x, \tau) = \exp\left(-\frac{\|y - x\|^2}{\tau}\right)$, *there exists a position embedding* $h(x, \tau) \in \mathbb{R}^n$ *with non-negative components such that* $\langle h(x, \tau), h(y, \tau)\rangle = q(y|x, \tau)$, *and each component* $h_i(x, \tau)$ *is a Gaussian function of* $x$ *with variance* $\tau/2$ *per dimension.*

*Proof.* The transition kernel $q(y|x, \tau) = \exp\left(-\frac{\|y - x\|^2}{\tau}\right)$ is a positive definite Gaussian kernel with variance $\tau/2$ (since $4\alpha\tau = \tau$ for $\alpha = 1/4$), consistent with the Gaussian tuning of place cells in open fields. We construct $h(x, \tau) \in \mathbb{R}^n$ directly as:

$$h_i(x, \tau) = c_i \exp\left(-\frac{\|x - \mu_i\|^2}{\tau}\right) \tag{35}$$

where $\mu_i \in \mathbb{R}^2$ are fixed anchor points (e.g., a uniform grid), $c_i > 0$ are constants, and the variance per dimension is $\tau/2$. The inner product is:

$$\langle h(x, \tau), h(y, \tau)\rangle = \sum_{i=1}^{n} h_i(x, \tau)h_i(y, \tau) = \sum_{i=1}^{n} c_i^2 \exp\left(-\frac{\|x - \mu_i\|^2 + \|y - \mu_i\|^2}{\tau}\right) \tag{36}$$

Rewrite the exponent:

$$\|x - \mu_i\|^2 + \|y - \mu_i\|^2 = \|x - y\|^2 + 2\left\|\frac{x + y}{2} - \mu_i\right\|^2 \tag{37}$$

so:

$$\langle h(x, \tau), h(y, \tau)\rangle = \exp\left(-\frac{\|x - y\|^2}{\tau}\right) \sum_{i=1}^{n} c_i^2 \exp\left(-\frac{2\left\|\frac{x + y}{2} - \mu_i\right\|^2}{\tau}\right) \tag{38}$$

For a dense set of $\mu_i$ (e.g., a grid with spacing $\ll \sqrt{\tau}$), the sum approximates a constant over a local region around $(x + y)/2$, as the terms $\exp\left(-\frac{2\left\|\frac{x + y}{2} - \mu_i\right\|^2}{\tau}\right)$ form a kernel density estimate. Set $c_i^2 = \frac{1}{n}$; as $n \to \infty$, the sum converges to a constant $C$ (proportional to the density of $\mu_i$), yielding:

$$\langle h(x, \tau), h(y, \tau)\rangle \approx C \exp\left(-\frac{\|x - y\|^2}{\tau}\right) \tag{39}$$

Adjust $C = 1$ by scaling $c_i$ (e.g., $c_i = \sqrt{\frac{1}{Cn}}$), ensuring $\langle h(x, \tau), h(y, \tau)\rangle = q(y|x, \tau)$. Each $h_i(x, \tau)$ is Gaussian with variance $\tau/2$ per dimension, and $h_i(x, \tau) \geq 0$, satisfying the theorem. $\square$

This open-field case not only validates our model against biological data but also serves as a baseline to explore how environmental structure (e.g., obstacles) perturbs these Gaussian properties, as observed in constrained settings [62, 63].

## F   Emergent Sparsity from Non-negativity and Orthogonality

Let $h(x, \tau) \in \mathbb{R}_+^n$ denote the non-negative representation of position $x \in \mathcal{X}$ at scale $\sqrt{\tau}$, and let

$$\langle h(x, \tau), h(y, \tau) \rangle = q(y \mid x, \tau) \tag{40}$$

where $q(\cdot \mid \cdot, \tau) \geq 0$ is a symmetric multi-step transition kernel (adjacency at scale $\sqrt{\tau}$). Define the (undirected) transition graph $\mathcal{G}_\tau = (\mathcal{X}, E_\tau)$ by

$$(x, y) \in E_\tau \quad \Longleftrightarrow \quad q(y \mid x, \tau) > 0 \tag{41}$$

Note that this transition graph is for $\tau$ steps instead of one step. Write $\text{supp}(v) = \{i \in [n] : v_i > 0\}$ for the support of a non-negative vector $v \in \mathbb{R}_+^n$, and put

$$S_i(\tau) = \{x \in \mathcal{X} : h_i(x, \tau) > 0\} \quad \text{(the active set of coordinate } i \text{ at scale } \tau) \tag{42}$$

Let $\deg_\tau(x)$ denote the degree of $x$ in $\mathcal{G}_\tau$, $\Delta(\tau) = \max_x \deg_\tau(x)$ the maximum degree, and

$$\rho(\tau) = \frac{1}{|\mathcal{X}|} \sum_{x \in \mathcal{X}} \deg_\tau(x) \tag{43}$$

the mean neighborhood size.

**Lemma 2** (Non-negativity + orthogonality $\Rightarrow$ disjoint support). *If $h(x, \tau), h(y, \tau) \in \mathbb{R}_+^n$ and $\langle h(x, \tau), h(y, \tau) \rangle = 0$, then*

$$\text{supp}(h(x, \tau)) \cap \text{supp}(h(y, \tau)) = \varnothing \tag{44}$$

*Proof.* By non-negativity,

$$0 = \langle h(x, \tau), h(y, \tau) \rangle = \sum_{i=1}^n h_i(x, \tau) \, h_i(y, \tau) \tag{45}$$

is a sum of non-negative terms. Hence each term vanishes: $h_i(x, \tau) \, h_i(y, \tau) = 0$ for all $i \in [n]$, and thus no coordinate is simultaneously positive in both vectors. $\qquad\square$

**Proposition 3** (Clique structure and sparsity bound). *Assume (40) and that*

$$q(y \mid x, \tau) = 0 \quad \text{whenever} \quad (x, y) \notin E_\tau \tag{46}$$

*Then for each coordinate $i \in [n]$:*

(i) *$S_i(\tau)$ induces a* clique *in $\mathcal{G}_\tau$; i.e., every pair $x, y \in S_i(\tau)$ is adjacent in $\mathcal{G}_\tau$*

(ii) *Consequently,*

$$|S_i(\tau)| \leq 1 + \min_{x \in S_i(\tau)} \deg_\tau(x) \leq 1 + \Delta(\tau) \tag{47}$$

(iii) *The average number of active coordinates per position obeys*

$$\frac{1}{|\mathcal{X}|} \sum_{x \in \mathcal{X}} \|h(x, \tau)\|_0 = \frac{1}{|\mathcal{X}|} \sum_{i=1}^n |S_i(\tau)| \leq \frac{n \, (1 + \Delta(\tau))}{|\mathcal{X}|} \tag{48}$$

*Moreover, since $\Delta(\tau) = \max_x \deg_\tau(x)$ and $\rho(\tau)$ is the mean degree, any a priori bound of the form $\Delta(\tau) \leq C \, \rho(\tau)$ (with graph-dependent constant $C \geq 1$) yields*

$$\frac{1}{|\mathcal{X}|} \sum_x \|h(x, \tau)\|_0 \leq \frac{n \, [1 + C \, \rho(\tau)]}{|\mathcal{X}|} \tag{49}$$

*Proof. (i)* Take any $x, y \in S_i(\tau)$ with $x \neq y$. Then $h_i(x, \tau) > 0$ and $h_i(y, \tau) > 0$, so by Lemma 2 they cannot be orthogonal. By the hypothesis $q(y \mid x, \tau) = \langle h(x, \tau), h(y, \tau) \rangle$, orthogonality occurs exactly when $(x, y) \notin E_\tau$. Therefore $(x, y) \in E_\tau$. Since the choice of $x, y$ was arbitrary, $S_i(\tau)$ induces a clique.

*(ii)* Fix any $x^* \in S_i(\tau)$. By *(i)*, every $y \in S_i(\tau) \setminus \{x^*\}$ must be adjacent to $x^*$, so $S_i(\tau) \subseteq \{x^*\} \cup \mathcal{N}_\tau(x^*)$, where $\mathcal{N}_\tau(x^*)$ is the neighborhood of $x^*$. Thus $|S_i(\tau)| \leq 1 + \deg_\tau(x^*)$. Minimizing over $x^* \in S_i(\tau)$ gives the first inequality; the second follows from $\deg_\tau(x^*) \leq \Delta(\tau)$.

*(iii)* The identity

$$\sum_{x \in \mathcal{X}} \|h(x,\tau)\|_0 \;=\; \sum_{i=1}^{n} |S_i(\tau)| \tag{50}$$

is a double-counting equality (both sides count the number of pairs $(x,i)$ with $h_i(x,\tau) > 0$). Using *(ii)* and summing over $i$ yields $\sum_i |S_i(\tau)| \leq n\,[1 + \Delta(\tau)]$, and dividing by $|\mathcal{X}|$ establishes (48). If an a priori comparison $\Delta(\tau) \leq C\,\rho(\tau)$ holds for the graph family under consideration, the final bound follows immediately. $\qquad\square$

**Geometric specialization.** If $\mathcal{X} \subset \mathbb{R}^m$ is $\eta$-separated (minimum inter-point distance $\geq \eta > 0$) and $\mathcal{G}_\tau$ is a geometric threshold graph

$$(x,y) \in E_\tau \quad \Longleftrightarrow \quad \|x - y\| \leq \sqrt{\tau} \tag{51}$$

then each closed neighborhood fits inside a ball $B(x, \sqrt{\tau})$ of radius $\sqrt{\tau}$. A standard packing argument implies

$$\deg_\tau(x) \;+\; 1 \;\leq\; C_m \left( \frac{\sqrt{\tau}}{\eta} \right)^m \tag{52}$$

for a constant $C_m$ depending only on the ambient dimension $m$. Consequently,

$$|S_i(\tau)| \;\leq\; C_m \left( \frac{\sqrt{\tau}}{\eta} \right)^m \qquad \frac{1}{|\mathcal{X}|} \sum_x \|h(x,\tau)\|_0, \;\leq\; \frac{d\,C_m}{|\mathcal{X}|} \left( \frac{\sqrt{\tau}}{\eta} \right)^m \tag{53}$$

Thus, as $\tau$ increases, the allowable clique size (hence the bound on average support) grows polynomially with the geometric volume of the $\sqrt{\tau}$-ball, quantitatively linking scale to reduced sparsity (broader fields).

**Corollary 4** (Scale-dependent localization). *Assume $\tau \mapsto \mathcal{G}_\tau$ is monotone in the sense that $\tau_1 < \tau_2$ implies $E_{\tau_1} \subseteq E_{\tau_2}$. Then $\Delta(\tau)$ and $\rho(\tau)$ are non-decreasing in $\tau$, and the upper bounds in Proposition 3 are non-decreasing in $\tau$, i.e., expected sparsity decreases with scale, corresponding to larger place fields.*

*Proof.* Monotonicity $E_{\tau_1} \subseteq E_{\tau_2}$ implies $\deg_{\tau_1}(x) \leq \deg_{\tau_2}(x)$ for all $x$, hence $\Delta(\tau_1) \leq \Delta(\tau_2)$ and $\rho(\tau_1) \leq \rho(\tau_2)$. Apply Proposition 3 (iii). $\qquad\square$

**Interpretation.** Lemma 2 enforces disjoint supports for non-adjacent locations; Proposition 3 shows that each coordinate $i$ can only support a *clique* of mutually adjacent positions and bounds the size of that clique by local neighborhood size. In geometric environments, the bound scales with the volume of a $\sqrt{\tau}$-ball, making the $\tau$–field-size relationship explicit. Corollary 4 then formalizes the empirical trend: smaller $\tau \Rightarrow$ higher sparsity (smaller fields); larger $\tau \Rightarrow$ lower sparsity (larger fields).

# G   Path Planning Properties

This appendix elaborates on the properties of the gradient-based navigation framework introduced in Section 2.7.

## G.1   Scale Transition Dynamics

The optimal scale $\tau^* = \arg\max_t \|\nabla_x q(x|y,\tau)\|$ adapts dynamically to the agent's distance from the goal, ensuring efficient navigation. In an open field, we formalize this selection with the following theorem.

**Theorem 5** (Optimal Scale Selection in an Open Field). *In an open field with a Gaussian transition kernel $p(x|y,\tau) = \frac{1}{4\pi\alpha\tau}\exp\left(-\frac{\|x-y\|^2}{4\alpha\tau}\right)$, where $\alpha = 1/4$, the optimal time scale $\tau^*$ that maximizes the gradient magnitude $\|\nabla_x q(x|y,\tau)\|$, with $q(x|y,\tau) = p(x|y,\tau)/\sqrt{p(x|x,\tau)\cdot p(y|y,\tau)}$, is:*

$$\tau^* = \frac{d^2(x,y)}{4\alpha} = d^2(x,y) \tag{54}$$

*where $d(x,y) = \|x-y\|$ is the Euclidean distance.*

*Proof.* The transition kernel is:

$$p(x|y,\tau) = \frac{1}{4\pi\alpha\tau}\exp\left(-\frac{d^2}{4\alpha\tau}\right), \quad d = d(x,y) \tag{55}$$

Normalization gives:

$$p(x|x,\tau) = p(y|y,\tau) = \frac{1}{4\pi\alpha\tau}, \quad \sqrt{p(x|x,\tau)\cdot p(y|y,\tau)} = \frac{1}{4\pi\alpha\tau} \tag{56}$$

Thus:

$$q(x|y,\tau) = \frac{p(x|y,\tau)}{\frac{1}{4\pi\alpha\tau}} = \exp\left(-\frac{d^2}{4\alpha\tau}\right) \tag{57}$$

The gradient is:

$$\nabla_x q(x|y,\tau) = \exp\left(-\frac{d^2}{4\alpha\tau}\right)\cdot\left(-\frac{2(x-y)}{4\alpha\tau}\right) = -\frac{x-y}{2\alpha\tau}\exp\left(-\frac{d^2}{4\alpha\tau}\right) \tag{58}$$

The magnitude is:

$$\|\nabla_x q(x|y,\tau)\| = \frac{d}{2\alpha\tau}\exp\left(-\frac{d^2}{4\alpha\tau}\right) \tag{59}$$

Maximize $f(\tau) = \frac{d}{2\alpha\tau}\exp\left(-\frac{d^2}{4\alpha\tau}\right)$. Compute:

$$\ln f(\tau) = \ln\left(\frac{d}{2\alpha}\right) - \ln\tau - \frac{d^2}{4\alpha\tau} \tag{60}$$

Differentiate:

$$\frac{\partial \ln f}{\partial \tau} = -\frac{1}{\tau} + \frac{d^2}{4\alpha\tau^2} = 0 \implies \tau = \frac{d^2}{4\alpha} \tag{61}$$

For $\alpha = 1/4$, $\tau^* = d^2$. The second derivative at $\tau^*$:

$$\frac{\partial^2 \ln f}{\partial \tau^2} = \frac{1}{\tau^2} - \frac{d^2}{2\alpha\tau^3}, \quad \text{at } \tau = \frac{d^2}{4\alpha}, \quad \frac{d^2}{2\alpha\tau} = 2, \quad \frac{\partial^2 \ln f}{\partial \tau^2} = -\frac{1}{\tau^2} < 0 \tag{62}$$

confirms a maximum. □

In an open field, $\tau^* \propto d^2(x,y)$, so $\tau^*$ decreases as the agent approaches the goal, focusing on finer spatial scales for precision.

## G.2 Properties of the Gradient Field

The gradient field $\nabla_x q(x|y,\tau)$ drives navigation. We highlight three properties ensuring a smooth, trap-free path to the goal $y$.

First, $p(x|y,\tau)$ satisfies the heat equation with reflecting boundary conditions:

$$\frac{\partial p(x|y,\tau)}{\partial \tau} = \alpha\nabla^2 p(x|y,\tau), \quad \alpha = 1/4, \quad \frac{\partial p(x|y,\tau)}{\partial n} = 0 \text{ on } \partial\Omega_{\text{obstacles}}. \tag{63}$$

For fixed $y$ and $\tau$, $p(x|y,\tau)$ is smooth in $x$, as the heat kernel (e.g., $p(x|y,\tau) = \frac{1}{4\pi\alpha\tau}\exp\left(-\frac{\|x-y\|^2}{4\alpha\tau}\right)$ in an open field) is infinitely differentiable [15]. It has a unique maximum at $x = y$.

Second, $p(x|x,\tau)$ is smooth in $x$ for fixed $\tau$. In an open field, $p(x|x,\tau) = \frac{1}{4\pi\alpha\tau}$ is constant, while in general, $p(x|x,\tau)$ varies smoothly due to the heat kernel's differentiability, reflecting the domain's geometry near obstacles.

Third, since the random walk is symmetric, $\nabla_x q(x|y,\tau) = \nabla_x q(y|x,\tau)$. The gradient field of $q(y|x,\tau)$ is smooth, as $q(y|x,\tau) = q(x|y,\tau)$ inherits the smoothness of $p(x|y,\tau)$, and has a unique maximum at $x = y$.

These properties ensure that $\nabla_x q(y|x,\tau)$ forms a smooth field with a unique maximum at the goal, producing a continuous, trap-free path toward $y$. This mirrors the hippocampus's efficient navigation, where place cells encode smooth, goal-directed trajectories [11].

### G.3 Planned Path vs. Shortest Path

The planned path follows the gradient $\nabla_x q(x|y,\tau)$, and $p(x|y,\tau)$ satisfies the heat equation with reflecting boundary conditions. We compare this path to the shortest (geodesic) path from $x$ to the goal $y$.

In an open field, the planned path is a straight line. The transition kernel is:

$$p(x|y,\tau) = \frac{1}{4\pi\alpha\tau}\exp\left(-\frac{\|x-y\|^2}{4\alpha\tau}\right), \quad \alpha = 1/4 \tag{64}$$

Since $p(x|x,\tau) = p(y|y,\tau) = \frac{1}{4\pi\alpha\tau}$, the normalization gives:

$$q(x|y,\tau) = \exp\left(-\frac{\|x-y\|^2}{4\alpha\tau}\right) \tag{65}$$

The gradient is:

$$\nabla_x q(x|y,\tau) = -\frac{x-y}{2\alpha\tau}\exp\left(-\frac{\|x-y\|^2}{4\alpha\tau}\right) \propto -(x-y) \tag{66}$$

Thus, $\frac{dx(t)}{dt} = \nabla_x q(x|y,\tau)$ follows a straight line from $x$ to $y$, matching the Euclidean shortest path [64].

For small $\tau$, $q(x|y,\tau)$ aligns with the geodesic distance:

$$p(x|y,\tau) \sim \frac{1}{(4\pi\alpha\tau)^{d/2}}\exp\left(-\frac{d_g^2(x,y)}{4\alpha\tau}\right) \tag{67}$$

where $d_g(x,y)$ is the geodesic distance [16]. Since $q(x|y,\tau) \propto p(x|y,\tau)$ in an open field, $\nabla_x q(x|y,\tau) \propto -\nabla_x d_g^2(x,y)$, ensuring the planned path follows the shortest route, even around obstacles.

For large $\tau$, $q(x|y,\tau)$ emphasizes global topological connectivity over local geometric details, often producing paths superior to the shortest. The transition matrix $P_\tau = P_1^\tau$ has spectral decomposition:

$$P_\tau = Q\Lambda^\tau Q^T \tag{68}$$

where $Q$ is orthogonal, and $\Lambda = \text{diag}(\lambda_1, \ldots, \lambda_n)$ contains eigenvalues $0 \leq \lambda_i \leq 1$ [57]. For large $\tau$, dominant eigenvalues ($\lambda_i \approx 1$) amplify global connectivity:

$$p(x|y,\tau) = [P_\tau]_{xy} \approx \sum_{i:\lambda_i\approx 1} \lambda_i^\tau Q_i(x)Q_i(y) \tag{69}$$

The eigenvectors $Q_i$ corresponding to $\lambda_i \approx 1$ encode topological features, such as major pathways and passage connectivity, invariant to deformations like stretching [57]. Thus, $\nabla_x q(x|y,\tau)$ prioritizes routes with high connectivity (e.g., wider corridors or multiple paths), potentially deviating from the shortest path to favor robust, flexible trajectories [10]. For example, in a maze, the planned path may choose a longer but more reliable route through a well-connected passage, avoiding narrow or risky shortcuts.

This topological focus aligns with neuroscience observations, where hippocampal place cells encode abstract connectivity (e.g., room layouts) over precise metrics, enabling robust navigation in complex environments [11, 9, 3]. Paths driven by topology are often "better" than the shortest, as they prioritize accessibility and adaptability, reflecting cognitive strategies in spatial tasks.

The planned path's alignment with the shortest path for small $\tau$ and its topological robustness for large $\tau$ mirror the hippocampus's ability to balance precision and global structure, ensuring efficient navigation across diverse environments.

### G.4 Boundary Layer Effects

Near obstacles, the gradient field $\nabla_x q(x|y, \tau)$ aligns parallel to boundaries, preventing collisions. This behavior arises from the reflecting boundary condition of the heat equation governing $p(x|y, \tau)$, with $\partial p(x|y, \tau)/\partial n = 0$ on $\partial\Omega_{\text{obstacles}}$. The condition ensures the normal component of $\nabla_x p(x|y, \tau)$ vanishes at boundaries, creating a flow parallel to obstacles, with strength modulated by the goal's position and time scale $\tau$.

### G.5 Diffraction-Like Patterns

Diffraction, in the context of our navigation framework, refers to the bending of the gradient field $\nabla_x q(x|y, \tau)$ around obstacles, such as corners or narrow passages, analogous to how waves bend around edges in optics or acoustics. This phenomenon arises because the transition probability $p(x|y, \tau)$, governed by the heat equation, diffuses probability mass through available paths, concentrating gradients toward openings like passages or around corners. These diffraction-like patterns guide trajectories through high-gradient regions, ensuring efficient navigation in complex environments.

The recursive nature of the random walk generates diffraction-like patterns:

$$p(x|y, \tau + 1) = \sum_z p(x|z, \tau) \, p(z|y, 1) \tag{70}$$

This convolution reflects the diffusion of probability from $y$ to $x$ through intermediate points $z$, allowing $p(x|y, \tau)$ to "bend" around obstacles by accumulating contributions from multiple paths [15]. Near a corner or passage, the heat kernel explores paths that wrap around the obstacle, creating a gradient field that converges toward navigable routes.

For a narrow passage of width $w$, the transition probability approximates the geodesic distance $d_g(x, y)$ through the passage. For small to moderate $\tau$:

$$q(x|y, \tau) \approx \exp\left(-\frac{d_g^2(x, y)}{4\alpha\tau}\right) \tag{71}$$

The gradient is:

$$\nabla_x q(x|y, \tau) \approx -\frac{1}{2\alpha\tau} \exp\left(-\frac{d_g^2(x, y)}{4\alpha\tau}\right) \nabla_x d_g^2(x, y) \tag{72}$$

Since $\nabla_x d_g^2(x, y)$ points along the geodesic path through the passage, $\nabla_x q(x|y, \tau)$ converges toward the opening, creating a funneling effect. The width of this convergence region scales with $\sqrt{\tau}$, as the heat kernel's diffusion spreads over a distance proportional to $\sqrt{4\alpha\tau}$. For small $\tau$, the gradient tightly focuses on the passage, ensuring precise navigation. For large $\tau$, the broader diffusion enables early detection of passages from greater distances, smoothing trajectories by integrating multiple nearby paths.

At sharp corners, diffraction-like patterns emerge similarly. The heat kernel accumulates probability along paths bending around the corner, creating a curved gradient field that guides the agent past the obstacle. Unlike passages, corners lack a single geodesic path, so the gradient field spreads, resembling optical diffraction patterns where intensity peaks near edges. This spreading is more pronounced for large $\tau$, as the heat kernel explores longer, circuitous routes.

These patterns ensure robust navigation in complex environments. The funneling effect through passages and curved trajectories around corners mirror hippocampal navigation, where place cells

encode paths that navigate mazes or avoid obstacles [11, 9]. The $\tau$-dependent scaling reflects the hippocampus's ability to adjust focus, balancing precision for small $\tau$ with global awareness for large $\tau$, enhancing adaptability in varied spatial contexts [16].

### G.6    Topological Invariance

Topological invariance ensures that navigation behavior, driven by the gradient field $\nabla_x q(x|y, \tau)$, remains consistent under continuous deformations (e.g., stretching, bending) that preserve the environment's connectivity, such as the number of passages or loops. This robustness mirrors the hippocampus's cognitive maps, which encode abstract spatial relationships invariant to physical distortions [3].

The transition probability $p(x|y, \tau)$ arises from a random walk on a discrete grid, with one-step transition matrix $P_1$. For time $\tau$, the transition matrix is:

$$P_\tau = P_1^\tau \tag{73}$$

with spectral decomposition:

$$P_\tau = Q \Lambda^\tau Q^T \tag{74}$$

where $Q$ is orthogonal ($Q^T Q = I$), and $\Lambda = \mathrm{diag}(\lambda_1, \ldots, \lambda_n)$ contains eigenvalues $0 \leq \lambda_i \leq 1$ [57]. The eigenvectors $Q_i$ and eigenvalues $\lambda_i$ encode topological features, such as connectivity (number of connected components), bottlenecks (e.g., passages), and cycles. The second eigenvalue ($\lambda_2$) quantifies global connectivity, with higher values indicating tighter bottlenecks, while eigenvectors capture local structures like loops [15].

For large $\tau$, dominant eigenvalues ($\lambda_i \approx 1$) amplify global topological features:

$$P_\tau \approx \sum_{i:\lambda_i \approx 1} \lambda_i^\tau Q_i Q_i^T \tag{75}$$

emphasizing coarse connectivity (e.g., major pathways between regions). The eigenvectors corresponding to $\lambda_i \approx 1$ represent low-frequency modes that partition the environment into connected regions, invariant to metric distortions like stretching a corridor. This is because $Q_i$ depends on the graph's adjacency structure, not precise distances, ensuring stability under deformations that preserve connectivity [57]. For example, stretching a corridor adjusts transition probabilities proportionally, but the eigenvectors $Q_i$ retain the same connectivity patterns, as they reflect the graph's topology.

The gradient field is $\nabla_x q(x|y, \tau)$, where $p(x|y, \tau) = [P_\tau]_{xy}$. Since $P_\tau$'s eigenvectors are topologically invariant, $\nabla_x q(x|y, \tau)$ preserves navigation paths (e.g., through passages) across equivalent environments. For large $\tau$, the dominance of $\lambda_i \approx 1$ ensures that $\nabla_x q(x|y, \tau)$ prioritizes global connectivity, guiding trajectories along major routes regardless of local geometry.

This invariance aligns with hippocampal navigation, where place cells encode connectivity (e.g., room layouts) over precise metrics, enabling robust path planning under environmental changes [9]. The focus on topology for large $\tau$, driven by eigenvectors tied to $\lambda_i \approx 1$, reflects cognitive maps that emphasize structural relationships, as observed in rodent navigation [11, 3].

### G.7    Hippocampal Preplay and Shortcut Detection

Our path planning framework discovers novel shortcuts using only local exploration, mirroring hippocampal preplay, where neural sequences predict unexplored paths [7, 8]. This process relies on efficient computation of transition probabilities and adaptive scale selection, without memorizing past successful paths.

The one-step transition matrix $P_1$ defines probabilities of moving between adjacent grid points. Multi-step transition matrices $P_\tau = P_1^\tau$ for scales $\tau = 2^k$ ($k = 1, 2, \ldots, K$) are computed via matrix squaring:

$$P_2 = P_1^2, \quad P_4 = P_2^2, \quad P_8 = P_4^2, \quad \ldots, \quad P_{2^k} = P_{2^{k-1}}^2 \tag{76}$$

requiring $\log_2 \tau$ operations.

The path planning algorithm adaptively selects $\tau^* = \arg\max_t \|\nabla_x q(x|y, \tau)\|$, guiding trajectories via $\nabla_x q(x|y, \tau^*)$. For regions $A$ and $B$ connected by an unexplored passage, local exploration

defines $P_1$. Matrix squaring yields $P_\tau$, and adaptive $\tau$-selection ensures $q(a|b, \tau^*) > 0$, discovering the shortcut without prior traversal.

This process—matrix squaring to compute $P_\tau$ and adaptive $\tau$-selection—embodies hippocampal preplay, predicting connectivity from local data, akin to place cell sequence pre-activation. The framework's efficiency and reliance on $P_1$ enhance its biological plausibility, supporting robust navigation.

# H    Theta Phase Modeling

We propose a local definition of theta phase that embeds spatial adjacency within the phase angle, capturing the observed precession range of 270° to 90° while enabling navigation across a tessellated environment. This formulation leverages an external 8 Hz theta rhythm for temporal pacing, where $t$ denotes real time during navigation, distinct from the random walk time scale $\tau$, and aligns with distributed hippocampal computation without requiring individual cells to explicitly encode spatial centers.

## H.1    From Inner Product Angle to Theta Phase

The emergent sparsity from non-negativity and orthogonality constraints endows each cell in $h(x, \tau)$ a localized place field, so that each cell $i$ has a place field centered at $\mu_i$. The place fields of all the cells tile the whole environment. At the current position $x(t)$ at real time $t$, the adjacency between $x(t)$ and each $\mu_i$ is measured by the angle between $h(x(t), \tau)$ and $h(\mu_i, \tau)$, and this angle can then be embedded as the theta phase.

Specifically, define the adjacency of the current position $x(t)$ to the field center of the $i$-th place cell at scale $\tau$:

$$a_i(t) = \langle h(x(t), \tau), h(\mu_i, \tau) \rangle \tag{77}$$

where $h(x(t), \tau) \in \mathbb{R}^n$ is the population embedding (firing rates of $n$ place cells) at position $x(t)$, and $h(\mu_i, \tau)$ represents the embedding at the center $\mu_i$, where

$$h_i(\mu_i, \tau) = \max_{x \in \mathcal{X}} h_i(x, \tau) \tag{78}$$

We interpret $h(\mu_i, \tau) = [w_{1i}, w_{2i}, \ldots, w_{ni}]$ as the recurrent connection weights from the population to neuron $i$, such that:

$$a_i(t) = \sum_{j=1}^{n} h_j(x(t), \tau) w_{ji} \tag{79}$$

representing the net recurrent input to neuron $i$. This corresponds to the normalized transition probability $q(x(t)|\mu_i, \tau)$, and in open fields (Appendix E), it follows a Gaussian profile:

$$a_i(t) = \exp\left(-\frac{\|x(t) - \mu_i\|^2}{\tau}\right) \tag{80}$$

The theta phase is defined as:

$$\phi_i(x(t), \tau) = \pi + \mathrm{sgn}\left(\frac{da_i(t)}{dt}\right) \cdot \left(\frac{\pi}{2} - \arccos(a_i(t))\right) \tag{81}$$

where: $\frac{da_i(t)}{dt} = \sum_{j=1}^{n} \frac{dh_j(x(t), \tau)}{dt} w_{ji}$ is the temporal rate of change of this recurrent input, reflecting movement toward ($> 0$) or away ($< 0$) from $\mu_i$ as encoded by the weights. $\arccos(a_i(t)) \in [0, \pi/2]$ is the angle between $h(x(t), \tau)$ and $h(\mu_i, \tau)$, and this angle is embedded into the phase, possibly by sinusoidal inhibitory mechanism [65, 66].

The phase progresses as:

1. At field entry ($a_i \to 0$, $\frac{da_i}{dt} > 0$): $\phi_i = \frac{3\pi}{2}$ (270°), late phase.

2. At field center ($a_i = 1$, $\frac{da_i}{dt} = 0$): $\phi_i = \pi$ (180°), mid-phase.

3. At field exit ($a_i \to 0$, $\frac{da_i}{dt} < 0$): $\phi_i = \frac{\pi}{2}$ (90°), early phase.

This does not require neuron $i$ to encode $\mu_i$ explicitly; rather, $h(\mu_i, \tau)$ emerges as a distributed representation within recurrent connectivity, learned through experience and computed collectively, with $\phi_i$ storing $a_i(t)$ as an angular signal.

### H.2 Implications for Navigation

This phase also enables navigation by tessellating the environment with $\{\mu_i\}$. Each $\mu_i$ anchors a tile, and $\phi_i$ signals adjacency: 180° indicates proximity to $\mu_i$, while 270° and 90° mark entry and exit. For a goal at $\mu_g$, navigating toward $\phi_g = \pi$ (max $a_g$) guides movement, with $\{\phi_i\}$ identifying intermediate $\mu_i$ to approach $\mu_g$. This discretizes continuous navigation into a phase-driven sequence over $\{\mu_i\}$, complementing our gradient-based approach with a recurrently-driven strategy. During rest, iterating $\phi_i$ at 8 Hz drives replay, with $\frac{da_i}{dt}$ reflecting virtual transitions, unifying coding and navigation.

By tessellating the environment with $\{\mu_i\}$ and guiding navigation via $\{\phi_i\}$, this approach unifies spatial representation, temporal coding, and path planning within the embedding framework, offering an elegant, biologically plausible mechanism rooted in recurrent connectivity and the random walk model.

## I Grid Cells Integration

### I.1 Grid Cells as a Conformal, Multi-Scale Basis

While place cells form the primary focus of our model, we establish a connection to grid cells in the medial entorhinal cortex. Grid cells exhibit hexagonal firing patterns across multiple scales [67, 9] and are thought to provide a universal metric that feeds into the more adaptive place cell system.

Let $g(x)$ be the vector formed by the firing rates of the population of grid cells at position $x$. The high dimensional $g(x)$ forms an embedding of 2D position $x$. The grid cell representation $g(x)$ provides an effective basis for learning the transformation matrix $W(\tau)$ to produce place cell embeddings $h(x, \tau) = W(\tau)g(x)$ that approximate $q(y|x, \tau)$. Its effectiveness stems from conformal, isotropic, and multi-scale properties of grid cells, aligning with the hippocampus's multi-resolution encoding [22].

While place cell representation $h(x, \tau)$ preserves the proximity within the environment, $g(x)$ preserves the local distance and has the property of conformal isometry.

*Conformal isometry* [68–71] is defined as:

$$\|g(x + dx) - g(x)\| = s\|dx\| + o(\|dx\|) \tag{82}$$

where $s$ is a scaling factor, and $o(\|dx\|)$ denotes higher-order terms vanishing as $\|dx\| \to 0$, ensuring local distance and angle preservation and isotropic scaling.

$s$ plays the role of metric. Grid cell population consists of multiple modules, with each module corresponding to a metric $s$, mirroring the multi-scale property of place cells.

### I.2 Implementation of Grid-to-Place Transformation

The place cell embedding is modeled as $h(x, \tau) = W(\tau)g(x)$, or more generally $h(x, \tau) = \text{Norm}(\text{ReLU}(W(\tau)g(x) + b(\tau)))$ where the transformation matrix $W(\tau)$ is learned to optimize the proximity-preserving property:

$$\langle W(\tau)g(x), W(\tau)g(y) \rangle \approx q(y|x, \tau) \tag{83}$$

The matrix $W(\tau)$ serves a dual purpose: (1) it captures the appropriate spatial scale by weighting the contribution of each grid cell module to match the place cell scale $\tau$, and (2) it adjusts for deviations from isotropy and isometry in grid cells. $g(x)$ facilitates fast adaptation to the environment in that learning $W(\tau)$ may be more efficient than learning $h(x, \tau)$ directly.

To further enhance adaptability, we can backpropagate gradients to $g(x)$. This allows the grid cell representation to deform in response to environmental constraints, simulating the dynamic remapping observed in biological grid cells [72].

This formulation provides a computational account of how place field embeddings adapt to environmental constraints based on a flexible metric provided by grid cells. The transformation matrix $W(\tau)$, combined with the learnable parameters of $g(x)$, effectively functions as a computational cognitive map, translating the multi-scale grid metric into place representations that reflect both the appropriate spatial scale and deviations from isometry and isotropy.

## J    Implementation Details

All the models were trained on a single NVIDIA A6000 GPU for 2,000 iterations with a learning rate of 0.001. For fine-tuning, we use 50 iterations with a 0.0005 learning rate. All models contain 500 cells. For batch size, we learn all combinations of different locations in the field for each iteration. The training time is less than 5 minutes.

We evaluate planning using two metrics: (1) Success, whether the agent reaches within 1 unit of the goal, and (2) Success weighted by inverse Path Length (SPL) [28]. For an agent's path $[x_1, \ldots, x_T]$ with initial geodesic distance $d_i$ for episode $i$ (as computed by baseline algorithms), we compute:

$$l_i = \sum_{\tau=2}^{T} \|x_\tau - x_{\tau-1}\|_2 \tag{84}$$

Then SPL for episode $i$ is defined as:

$$\text{SPL}_i = \text{Success}_i \cdot \frac{d_i}{l_i} \tag{85}$$

We report SPL as the average of $\text{SPL}_i$ across all episodes. In Table 1, we treat Bug algorithm with oracle as the baseline and in Table 2, our method serves as the baseline.

## K    Additional Experiment Results

### K.1    Additional Path Planning Results in Four-Room Environment

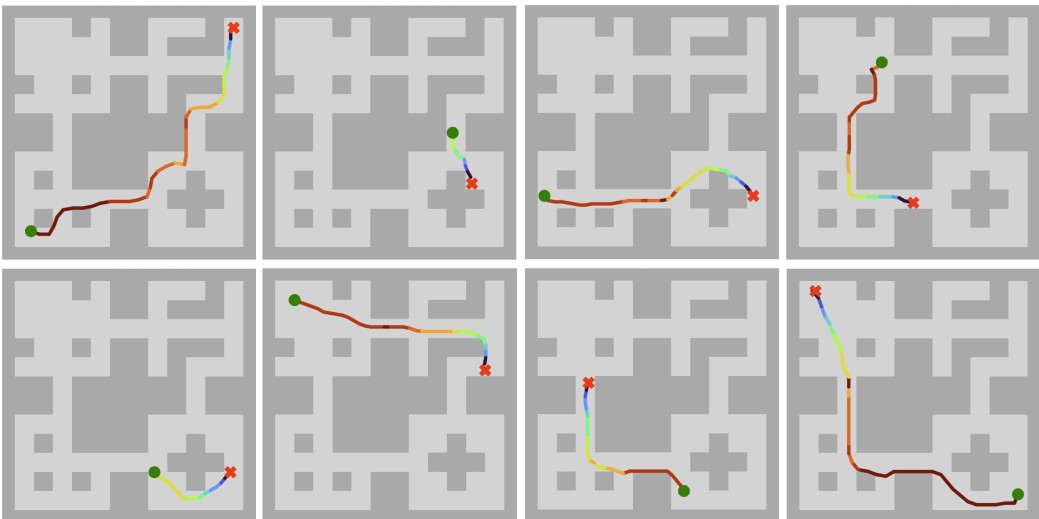

Figure 3: Successful navigation across eight randomly sampled start-goal configurations in the four-room environment.

To further demonstrate the robustness of our path planning framework, here we provide additional navigation trials in the four-room environment with randomly sampled start and goal locations.

Figure 3 shows eight scenarios spanning diverse spatial configurations. In all cases, our gradient-based path planning with adaptive scale selection successfully generated smooth, obstacle-avoiding trajectories that reached the goal. These results, combined with the quantitative evaluation in Table 1 showing 100% success rate across 50 random trials, demonstrate that our model achieves reliable navigation across diverse start-goal configurations even in complex multi-room environments.

## K.2 Remapping

We evaluated our model's ability to adapt to bidirectional environmental modifications in the four-room environment (our most complex tested scenario). In addition to the obstacle removal experiment shown in Figure 2 (D), we conducted the complementary scenario: initially, rooms were connected allowing direct northward navigation; we then added an obstacle blocking this direct pathway and fine-tuned the model with 100 iterations at learning rate 0.01. The agent successfully adapted, discovering alternative routes through adjacent rooms to reach the target (see Figure 4). Together, these experiments demonstrate topology-dependent remapping – a well-documented phenomenon where place fields reorganize in response to changes in spatial connectivity. By both removing obstacles (creating novel shortcuts) and adding obstacles (blocking direct paths), we show that minimal fine-tuning enables the position embeddings to adapt to environmental changes that alter topological connectivity, whether exploiting new connections or navigating around new barriers. This bidirectional adaptability aligns with hippocampal responses to environmental modifications.

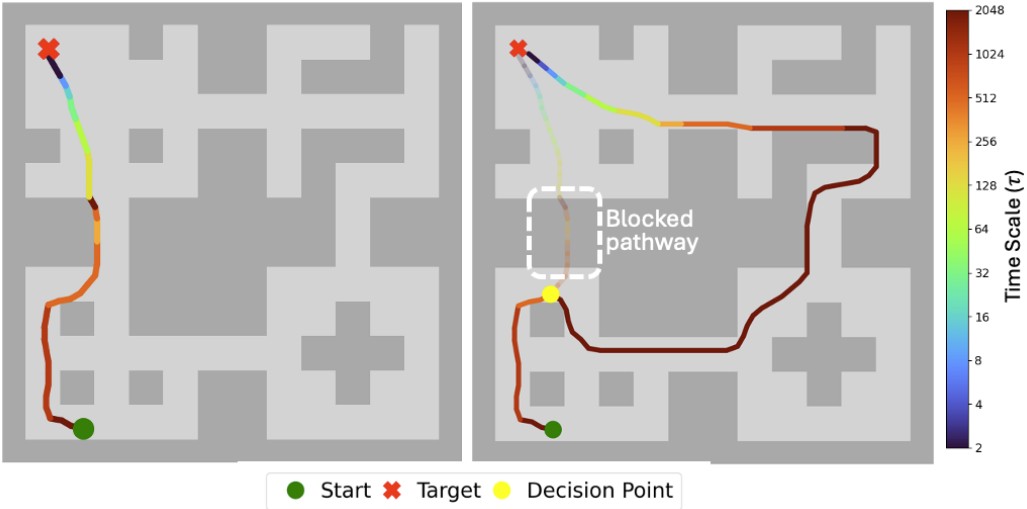

Figure 4: Remapping and path adaptation in response to added obstacles in the four-room environment.

