# OpenReview forum: "Place Cells as Multi-Scale Position Embeddings: Random Walk Transition Kernels for Path Planning"
_NeurIPS.cc/2025/Conference — NeurIPS 2025 poster_

### Official Review · Reviewer_GgX7 · 2025-06-05

**Clarity:** 3
**Significance:** 3
**Originality:** 3
**Rating:** 4
**Confidence:** 3

**Summary:**

This paper presents a new computational model for hippocampal navigation based on position embedding of multi-time random walk transition kernels. It is an interesting contribution to computational neuroscience, especially the community of spatial navigation. The paper is well written and easy to follow. Experimental results on maze simulation are limited (Figure 2 only) but interesting and convincing to some degree. Overall, I think this paper is at the borderline between 3 (WR) and 4(WA). I can go either way based on other reviewers' ratings and the authors' rebuttal.

**Questions:**

1. Can the proposed model be extended for goal-reaching policy with obstacles? In a classical study [1], geodesic distance navigation was demonstrated for the presence of obstacles. It will be interesting to show your model can achieve something similar.

2. In Sec. 2.4, why do you only consider matrix squaring instead of matrix multiplication - e.g., P_{m+n}=P_m\timesP_n?

3. Does path planning via adaptive gradient following have anything to do with the principle of least resistance? In [1], it has been shown how the minimum distance navigation can be achieved by the principle of least resistance.

4. How is the proposed model related to [2]? Have you try other environmental modifications such as adding an obstacle instead of opening a shortcut?

[1] Muller, R. U., Stead, M., & Pach, J. (1996). The hippocampus as a cognitive graph. The Journal of general physiology, 107(6), 663-694.
[2] Whittington, J. C., McCaffary, D., Bakermans, J. J., & Behrens, T. E. (2022). How to build a cognitive map. Nature neuroscience, 25(10), 1257-1272.

**Ethical Concerns:**

["NO or VERY MINOR ethics concerns only"]

**Final Justification:**

I am satisfied with authors' rebuttal and willing to increase my rating by one grade.

**Limitations:**

Yes. The paper has considered both theta-phase procession and grid cell integration as major limitations of this work.

**Quality:**

3

**Strengths And Weaknesses:**

+ The navigation and planning model presented in this work is interesting and has its novelty. Path planning via adaptive gradient flowing seems to be consistent with the principle of least resistance (though not explicitly discussed in the paper, please see my question below).
+ The remapping experiment in Fig. 2D is interesting and thought-provoking. It shows that the model can manage the opening of new pathway (i.e., to discover a shortcut).

- The model's simplicity makes me wonder about its capability for handling more sophisticated environmental settings when geodesic distance is involved (e.g., what about the optimal path is concave?).
-The reported experimental results are limited. I was hoping that more simulation results will be included in Appendix. But no additional results can be found there.
-I am a bit skeptical about the remapping experiment in 3.2.4. There are different ways of defining remapping in hippocampal study - here authors seem to only consider one "analogous to the remapping phenomenon".

---

> ### Author Rebuttal · Authors · 2025-07-31
>
> We thank you for recognizing our model as an interesting and novel contribution to computational neuroscience.
>
> ### Weaknesses
>
> >**1. Model Simplicity**
>
> **Geodesic distance and complex environments:** Thank you for this important concern! Our model handles sophisticated environments, including concave optimal paths, through its foundation in symmetric random walks and heat equation connections. The key insight is that transition probabilities $q(y|x,t)$ are not hand-crafted but emerge from random walk dynamics that naturally respect environmental constraints.
>
> **Theoretical foundation for complex geometries:** As detailed in Section 2.2 and Appendix C, our discrete random walk converges to the heat equation with reflecting boundary conditions. Crucially, Varadhan's formula (Appendix C.4) establishes that for small $t$, our learned $q(y|x,t)$ approximates $exp(-d_g^2(x,y) / (4\alpha t))$, where $d_g(x,y)$ is the *geodesic distance* through the environment. This means our model implicitly encodes true shortest paths that navigate around obstacles, even when these paths are highly concave.
>
> **Empirical evidence:**
> Our experiments can demonstrate this capability:
> - The "diffraction-like patterns" in Figure 2B show how gradient fields bend around corners and through passages
> - In the S-shaped and Four-room mazes (Figure 2A), our model achieves 100% success with superior efficiency (SPL > 1.0) despite inherently concave optimal paths
> - The boundary layer effects (Appendix E.4) ensure obstacle-parallel flows for complex navigation.
>
> **Multi-scale robustness:**
> Adaptive scale selection automatically adjusts resolution—coarse scales for global planning around large obstacles, fine scales for local precision—making our approach effective for concave paths requiring both global awareness and local navigation.
>
> **Additional experiment validation:**
> Following your advice, we tested our model on the concave environment from Figure 10 of Muller et al. [1], achieving 100% success rate.
>
> >**2. Additional results**
>
> We sincerely thank the reviewer for this valuable feedback. Our primary aim was to demonstrate the fundamental properties of our framework across key dimensions: biological plausibility through place cell characteristics (field size distribution, adaptability, remapping), and functional capabilities in robust path planning. We conducted experiments across multiple environment types. We selected the four-room maze as our most complex scenario because it presents multiple navigation challenges—hierarchical structure, bottlenecks, long-range planning, and multi-scale reasoning—effectively demonstrating our model's capabilities while maintaining biological plausibility.
>
> We agree that broader experimental validation would strengthen our work. In response to your feedback, we will significantly expand the experimental section to include additional complex maze configurations with varying geometric properties (concave).
>
> >**3. Remapping**
>
> We appreciate the opportunity to clarify our remapping experiment. The reviewer correctly notes that hippocampal remapping encompasses diverse phenomena (global, partial, rate remapping). In Section 3.2.4, we specifically modeled topology-dependent remapping—where removing obstacles creates novel shortcuts, introducing minimal physical but significant topological changes. This well-documented remapping type occurs when place fields reorganize to reflect new spatial relationships. Our model demonstrates that minimal fine-tuning enables embeddings to adapt and utilize new shortcuts, aligning with hippocampal responses to novel pathways. While we don't model all remapping forms, our experiment addresses a biologically relevant aspect highlighting topological adaptability. We will clarify this specific focus in our revision.
>
> ### Questions
> >**1. Goal-reaching with obstacles**
>
> Thank you for this question! Our model inherently handles goal-reaching with obstacles through Varadhan's formula, which ensures geodesic distance encoding even in complex environments. Our experimental validation demonstrates this capability: the U-shaped environment resembles Figure 15(A) in [1], while our four-room maze represents even greater complexity. Section 3.2 shows 100% success rates with superior efficiency (SPL > 1.0), indicating optimal geodesic navigation. We will add more classical environments from [1] to validate this capability further.
>
> >**2. Matrix multiplication**
>
> We consider matrix squaring ($P_{2t}=P_t^2$) as a primary method for two main reasons, highlighting both computational efficiency and biological plausibility:
>
> * **Computational Efficiency for a Scale Hierarchy:** Matrix squaring is exceptionally efficient for building a *multi-scale hierarchy* where the scales are powers of 2 (e.g., $t=2^k$). To compute $P_{2^k}$ from $P_1$, it only requires $k = \log_2(2^k)$ matrix multiplications. For example, to get $P\_{256}$ from $P\_1$ requires only 8 matrix multiplications ($P_2, P_4, P_8, \dots, P_{256}$). This is much faster than computing $P\_t$ as $P_1 \times P_1 \times \dots \times P_1$ ($t$ times).
> * **Biological Analogy to Preplay:** This efficient generation of global representations from local transitions enables hippocampal "preplay-like" path planning without memorizing trajectories. We hypothesize this occurs through rapid, iterative computation in recurrent circuits, where each "squaring" step propagates over conceptual distance, allowing offline shortcut discovery that mirrors preplay's predictive nature.
>
> We agree with you that we can also employ $P_{m+n}=P_m\times P_n$, and we mention in our paper that "one can design any discrete sequence $\mathcal{T}={t_k,k=1,...,K}$, and calculate $P_{t}$ using matrix multiplication".
>
> >**3. Least resistance**
>
> Thank you for suggesting this really interesting work! Indeed there is a strong conceptual connection between our adaptive gradient following and the "principle of least resistance" described in [1].
>
> In Muller et al. [1], the "least resistance" refers to the path through a "cognitive graph" that minimizes accumulated "resistance," where resistance might be related to synaptic weights or connection strengths, effectively finding the path of strongest connectivity or lowest cost. In our framework, the transition kernel $q(y|x,t)$ (and its unnormalized version $p(y|x,t)$) measures the "spatial adjacency" or "connectivity strength" between locations $x$ and $y$ at a given scale $t$. Path planning via adaptive gradient ascent on $q(y|x,t)$ (i.e., moving in the direction of maximal increase in transition probability towards the goal) means we are effectively seeking the path where there is the "most probable" or "easiest" flow from the current location towards the target. The gradient $\nabla_x q(x|y,t)$ acts like a "force field" guiding the agent. Paths following this gradient correspond to areas of high "connectivity" or "flow," which can be interpreted as paths of least resistance to reach the goal at that particular scale. Our adaptive scale selection refines this: small $t$ prioritizes shortest geodesic paths (least physical resistance), while large $t$ emphasizes global topological connectivity (least topological resistance via accessible corridors). This multi-scale approach aligns with cognitive maps integrating metric and topological properties, making our method a manifestation of least resistance principles adapted to multi-scale representations.
>
> >**4. Regarding Whittington et al. (2022)[2]**
>
> We thank the reviewer for highlighting the work by [2], which presents a framework showing how RNNs trained with SR objectives can learn cognitive maps. Both approaches share the goal of providing computational accounts of cognitive maps. While complementary, several key distinctions exist:
>
> * **Mathematical Foundation:** Our model is explicitly grounded in the mathematics of *random walks, heat diffusion, and spectral graph theory*. This allows for direct connections to established concepts like geodesic distance and provides a principled mechanism for multi-scale representation via the time parameter $t$.
> * **Explicit Multi-Scale Kernels:** While Whittington et al. implicitly learn multi-scale representations, we explicitly construct distinct transition kernels $P_t$ for various $t$ values, directly linking to dorsoventral scaling observations.
> * **Representational Format:** Our framework proposes place cells as position embeddings $h(x,t)$ derived from the transition kernels, with an emphasis on their inner product approximating transition probabilities $\langle h(x,t), h(y,t) \rangle = q(y|x,t)$. While Whittington et al. also use population codes, our specific inner product formulation and its connection to spectral decomposition and non-negative matrix factorization (via Horn's theorem) provide a distinct mathematical basis.
> * **Path Planning Mechanism:** Our path planning relies on adaptive gradient ascent on these multi-scale transition probability fields, which naturally produce robust navigation properties.
>
> **Experiments with environmental modifications:** Thank you for this suggestion. Our current remapping experiment (Section 3.2.4) focused on shortcut discovery to demonstrate cognitive map flexibility, but we have also tested obstacle addition. In the four-room environment, we added obstacles blocking direct pathways and fine-tuned the model. Initially, rooms were connected as shown in the lower panel of Figure 2(D), allowing direct northward navigation. The agent successfully adapted by discovering alternative routes through adjacent rooms when adding an obstacle. This demonstrates bidirectional adaptability—both exploiting new connections and navigating around obstacles. We can include this experiment in the revision for a more complete picture of adaptability.
>
> *Thank you again for your thoughtful review and constructive feedback. We are committed to addressing your concerns and will revise our paper accordingly.*

---

> > ### Comment · Reviewer_GgX7 · 2025-08-06
> >
> > Thank the authors for carefully answering my questions. I am mostly satisfied with your clarifications. Therefore, I have raised my rating by one grade. Unfortunately, due to some personal matters, I cannot engage in more interaction with the authors at this point. This paper has received the highest rating in the pool of papers assigned to me this year. I wish authors good luck with this submission!

---

> > > ### Author Response · Authors · 2025-08-07
> > >
> > > Thank you so much for your insightful review! We are committed to revising our paper based on your valuable comments and suggestions.

---

### Official Review · Reviewer_xpWW · 2025-06-24

**Clarity:** 4
**Significance:** 3
**Originality:** 2
**Rating:** 5
**Confidence:** 4

**Summary:**

This paper introduces a novel and elegant framework for modeling hippocampal place cells and spatial navigation. The central thesis is to reconceptualize a population of place cells as a single vector embedding, $h(x,t)$, for each location $x$ and time scale $t$. The key innovation is that the inner product of these embeddings, $⟨h(x,t),h(y,t)⟩$, is trained to approximate the normalized transition probability, $q(y∣x,t)$, of a symmetric random walk on the environment's manifold.

**Questions:**

The paper's formulation based on $P_t​=P_1^t​$ bears a strong resemblance to the SR, which is based on $∑(γP1​)k$. Could the authors provide a detailed discussion in the main text on the theoretical and practical relationship between their framework and SR? Is it more accurate to frame the contribution as a novel, scalable, factorized representation of a multi-scale transition kernel, rather than a wholly separate paradigm?

The learning objective in Eq. 8 appears to have a complexity of at least $O(N^22)$, which could be prohibitive for large environments. Could the authors clarify the computational complexity of the training phase and discuss whether any sampling or approximation techniques were used or would be necessary for scaling up?

The paper claims biological plausibility. While the resulting representations are plausible, the computational mechanisms are more abstract. Could the authors comment on the biological plausibility of the learning algorithm (gradient descent with AdamW) and, in particular, the matrix squaring operation ($P_{2t}​=P_t^2​) as a neural computation for replay?

The path planning relies on the gradient of the continuous kernel, which is constructed via bi-linear interpolation. This results in a $C^0$ continuous field whose gradient is discontinuous. Have the authors investigated the effect of these discontinuities on path quality, and would a smoother interpolation scheme (e.g., bicubic) yield better results?

**Ethical Concerns:**

["NO or VERY MINOR ethics concerns only"]

**Final Justification:**

The authors have significantly improved the paper and addressed all the concerns raised.

**Limitations:**

yes

**Quality:**

3

**Strengths And Weaknesses:**

The framework builds a rigorous bridge from a simple, discrete random walk to a sophisticated, continuous navigation policy. The derivation connecting the discrete walk to the continuous heat equation with reflecting boundary conditions is a classic and powerful approach from statistical physics, giving credibility to the model.

The use of Varadhan's formula to connect the short-time heat kernel to the squared geodesic distance ($d_g^2(x,y)$) provides a rigorous mathematical justification for why the learned transition probabilities encode meaningful spatial information, especially for fine-scale navigation. Moreover, to identify that neural firing rates must be non-negative and invoke Horn's theorem, providing existence of a non-negative matrix factorization for the symmetric, non-negative transition kernel, is cool.

The reported SPL values, which significantly exceed 1.0 in complex mazes like the S-shape and Four-room environments, demonstrate that the model finds substantially more efficient paths than the baseline.

The paper claims computational efficiency, primarily highlighting the matrix squaring step for inference. However, the training phase appears to be a potential bottleneck. The quadratic scaling could become prohibitive for the larger, more realistic environments where the model's multi-scale nature would be most beneficial.

The model learns embeddings on a discrete lattice and then uses bi-linear interpolation to create a continuous map for gradient-based planning. While a practical choice, bi-linear interpolation results in a function that is only $C^0$. This contradicts the claim of a perfectly "smooth gradient field". The impact of this choice on path quality should be discussed.

There appears to be a minor inconsistency between the theoretical derivation in Appendix C, which assumes a 4-neighbor random walk , and the main experiments, which use an 8-neighbor walk.

Visualizations are good.

The main text could benefit from more intuitive explanations to bridge the different concepts from statistical physics, spectral graph theory, computational neuroscience, and optimization. For example, a more gradual introduction to how the eigenvalues of the transition matrix relate to "topological connectivity".

The discussion of related work is put in Appendix A, which weakens the paper's main narrative by failing to immediately situate its key contributions. In particular, the distinction from the very closely related Successor Representation (SR) framework is not made in the main body of the paper.

The paper's primary significance lies in its potential as a unifying theory. It elegantly connects a simple, low-level process (a random walk) to the emergent, multi-scale, and adaptive properties of a high-level cognitive function (spatial navigation via a cognitive map). The model proposes concrete, computationally tractable mechanisms for phenomena observed in the hippocampus. Adaptive multi-scale representation could have significant potential for robotics.

A significant limitation is that the framework relies on pre-computing the transition matrix $P_1$ for the entire environment, and then computing all $P_t$ via matrix squaring before any planning can occur. This is an "offline" process, limiting its significance for modeling online learning in animals, which build maps through active exploration, and for robotic applications in dynamic or unknown environments.

The model operates on a discrete lattice where adjacency is pre-defined. It does not address the problem of how such a representation would be built from raw sensory inputs (e.g., vision).

The core formulation, $⟨h(x,t),h(y,t)⟩≈q(y∣x,t)$, is an original way to frame the population code of place cells. It moves beyond modeling individual firing fields to modeling the geometric structure of the entire transition kernel in an inner product space. Using the time parameter $t$ of a single random walk process to generate a continuous hierarchy of spatial scales is novel. The "diffusion-on-the-graph" approach provides a direct and interpretable way to incorporate environmental structure into the planning process.

The work does not sufficiently differentiate from the SR framework which models spatial representations based on powers of the transition matrix T (analogous to $P_1$). The SR matrix, often defined as $M=(I−γT) −1 =∑ _{k=0}^∞ (γT)^k$, represents the expected discounted future occupancy of states. The paper's transition kernel is $P_t =P_1^t$. Both are fundamentally based on powers of the transition matrix, and the paper's time parameter $t$ acts analogously to the SR's discount factor $γ$ in controlling the predictive horizon. While the authors' formulation using an inner product embedding is novel, the underlying mathematical object (powers of the transition matrix) is not. The paper fails to discuss this deep connection in the main text, leaving the reader to question whether this is a fundamentally new paradigm or an elegant and powerful reformulation of the same core principle as SR. This must be addressed.

The true novelty may lie not in proposing a new computational object (versus SR), but in proposing a new representational format. The learning objective is essentially a low-rank non-negative matrix factorization of the kernel matrix $Q_t$. Therefore, the contribution could be framed as a scalable, factorized representation of a multi-scale transition kernel, which has strong parallels to the SR but offers a distinct representational advantage. Clarifying this would strengthen the paper's claims of novelty.

---

> ### Author Rebuttal · Authors · 2025-07-31
>
> We are grateful that you found our framework novel and elegant.
>
> ### Weaknesses
>
> >**1. Computational Efficiency and Training Bottleneck**
>
> Our claim of computational efficiency primarily refers to the *inference* phase (path planning) and the rapid generation of multi-scale kernels via matrix squaring. We acknowledge that the training phase, which sums over all pairs of locations, has a complexity of $O(N^2)$, where $N$ is the number of lattice points. While feasible for our tested environments (40×40 grid, training under 5 minutes), this would be a bottleneck for much larger maps.
>
> To address this, our framework supports adaptive grid resolution tied to planning scale: coarse grids for long-range planning, fine grids for short-range precision. This multi-resolution strategy aligns with our model's multi-scale nature and manages training complexity. Future work will explore adaptive grids with sampling techniques like mini-batching for enhanced scalability.
>
> >**2. $C^0$ Continuity and the Gradient Field**
>
> The reviewer correctly observes that bi-linear interpolation yields a $C^0$ continuous function, whose true gradient is indeed discontinuous at grid boundaries. Our use of "smooth" was intended to be qualitative, reflecting that the resulting paths are not erratic.
>
> Specifically, this theoretical non-differentiability poses no practical problem for our path planning algorithm. In our numerical implementation, we approximate the gradient using a **finite difference method**. We evaluate $q(y∣z,t)$ at multiple candidate next positions $z\in∂(x)=\\{x+\Delta r(cos⁡θ,sin⁡θ)\\}$ around the current location $x$, where $\theta$ is discretized into equally spaced directions, and select the position that maximizes $q(y∣z,t)−q(y∣x,t)$ (Equation 9). This calculation only requires the function to be defined at these points; it does not require the existence of a continuous infinitesimal gradient.
>
> Therefore, our planner can still produce smooth, effective trajectories. We will clarify this point in the revised manuscript.
>
> >**3. Inconsistency in Random Walk Definition**
>
> We thank the reviewer for accurately identifying a minor discrepancy. Appendix C's theoretical derivation uses a 4-neighbor random walk for simplicity, while experiments use an 8-neighbor walk. The core connection to a diffusion process and geodesic distance holds for any symmetric random walk, with the specific diffusion coefficient $\alpha$ varying based on the neighborhood. We will clarify this in Appendix C, noting that the choice of neighborhood affects $\alpha$ but the theoretical properties of the heat equation connection remain.
>
> >**4. Intuitive Explanations for Concepts**
>
> We agree that clearer intuitive explanations are needed. We will revise the Introduction and Method sections to provide better conceptual build-up. Specifically, we'll explain that dominant eigenvalues capture "slow-changing" modes of the random walk that reveal global environmental structure—the topological layout of rooms and pathways that remains invariant under metric distortions. This demonstrates how our model naturally leverages random walk properties for robust navigation.
>
> >**5. Related Work in Appendix A**
>
> We agree with the reviewer. Placing related work in an appendix was a submission choice that indeed weakened the narrative. We will move a condensed, yet comprehensive, "Related Work" section to the main body of the paper, likely after the Introduction. Crucially, this section will explicitly introduce and differentiate our framework from the Successor Representation (SR), a point elaborated in response to comment 8.
>
> >**6. Offline Process Limitation**
>
> We are grateful for the reviewer's recognition of our paper as a unifying theory. The reviewer correctly notes our current framework is "offline," pre-computing $P_1$ for the entire environment. However, our model does enable "preplay-like path planning" through matrix squaring, which efficiently computes global transitions ($P_t$​) from local rules ($P_1$​) without memorizing trajectories—analogous to hippocampal preplay of novel shortcuts. Future work will address incremental $P_1$ construction and dynamic updates for online map building and adaptation.
>
> >**7. Lack of Sensory Input Integration**
>
> The reviewer correctly notes our model assumes a pre-defined discrete lattice, a limitation we acknowledge in Section 4. Our focus was on computational principles linking random walks to hippocampal representations, abstracting away perception. Future work will integrate sensory processing by learning $P_1$ from observations or deriving $h(x,t)$ directly from sensory features.
>
> >**8. Differentiation from Successor Representation (SR)**
>
> We thank the reviewer for this insightful observation about the connection to SR. Both frameworks build on the powers of transition matrices, situating our work within the broader literature. We will significantly revise the "Related Work" section to clarify this:
>
> 1. **Shared Foundation**: Both frameworks root spatial representations in transition matrices.
> 2. **SR's Discounted Sum**: SR ($M = \sum_{k=0}^\infty (\gamma T)^k$) represents expected discounted future occupancy, with $\gamma$ setting the predictive horizon.
> 3. **Our Multi-Time Transition Kernel**: Our approach uses discrete powers, $P_t = P^t$, where $t$ explicitly defines a "time step" or "spatial scale" ($\sqrt{t}$). We model probabilities at *distinct time horizons*, not a discounted sum.
> 4. **Analogy of $t$ and $\gamma$**: While $t$ and $\gamma$ both control predictive horizons, $t$ directly relates to a *physical time scale* of a random walk, aligning with distinct dorsoventral place field scaling.
>
> **Key Distinctions and Novel Contributions**
> 1. **Biologically-Constrained Factorization with Emergent Sparsity**:
> Our primary novelty lies in learning non-negative vector embeddings $h(x,t)$ such that $\langle h(x,t), h(y,t) \rangle = q(y|x,t)$, which is a biologically-constrained matrix factorization of the transition kernel.
> Crucially, we show that orthogonality + non-negativity automatically induces sparsity: When $q(y|x,t) \rightarrow 0$ for distant locations, the constraint $\langle h(x,t), h(y,t) \rangle = 0$ combined with $h(x,t), h(y,t) \geq 0$ forces disjoint support sets—explaining why place cells are naturally sparse and localized without explicit regularization.
> 2. **Multi-Scale Architecture and Adaptive Navigation**: Unlike SR's single discount factor, we explicitly model multiple scales $t \in {2^k}$ simultaneously via efficient matrix squaring ($P_{2t} = P_t^2$), directly mirroring hippocampal dorsoventral organization. Our adaptive gradient ascent selects optimal $t^*$ at each navigation step, enabling smooth, trap-free trajectories.
>
> The reviewer's insightful comments help clarify our contribution: we provide a scalable, biologically plausible realization of multi-scale transition kernels that bridge abstract SR principles with concrete neural population dynamics. The geometric insight that orthogonality + non-negativity = sparsity offers a new theoretical understanding of why spatial representations are naturally sparse and localized.
>
> We will incorporate proper SR citations and discussion in our revision.
>
> ### Questions
>
> >**1. Relationship to Successor Representation (SR)**
>
> As detailed in our response to reviewer Weakness 8, we will add a dedicated "Related Work" discussion. We will clarify that both frameworks derive spatial representations from transition matrices—SR uses discounted sums while we use discrete powers for multi-scale probabilities. Our core novelty lies in the representational format (non-negative embeddings $h(x, t)$) and explicit multi-scale hierarchy with adaptive navigation, offering a biologically interpretable paradigm.
>
> >**2. Computational Complexity of Training**
>
> See our response to Weakness 1. While the training is feasible for current environments, this scales quadratically. We will acknowledge this limitation and discuss that future work will explore sampling techniques (e.g., mini-batching pairs of $(x,y)$ during optimization) to reduce the complexity per iteration to $O(B)$, where $B$ is the batch size, making it scalable to much larger environments.
>
> >**3. Biological Plausibility of Mechanisms**
>
> While AdamW is an abstract optimization algorithm used to demonstrate learnability, biological learning mechanisms (e.g., Hebbian-like plasticity) may perform analogous optimizations. For matrix squaring ($P_{2t} = P_t^2$), which computes multi-step transitions from local rules, we propose that this could be mediated by recurrent neural circuits. Repeated, rapid "spreading activation" within hippocampal-entorhinal networks could effectively simulate multi-step transitions, akin to how the brain "preplays" and discovers novel shortcuts offline. This is an interpretation of the function that such neural circuits could perform, rather than a literal "matrix squaring" computation.
>
> >**4. Effect of $C^0$ Continuity on Path Quality**
>
> See our response to Weakness 2. Our path planner does not compute an analytical gradient but a **finite difference approximation**, taking a small but non-infinitesimal step to find the direction of ascent. This numerical approach effectively sidesteps the issue of discontinuities. At every point, including the boundaries between grid cells, the planner can evaluate the transition probability $q(y∣z,t)$ at each candidate position and determine the next step.
>
> Because our method does not rely on true differentiability, we did not observe any degradation in path quality attributable to these discontinuities. Our "near-optimal trajectories" validate this robustness. While smoother interpolation schemes remain theoretically interesting, our approach proves effective for high-quality navigation. We will clarify our numerical approach in the revision.
>
> *Thanks for your detailed feedback. We will incorporate all your insightful comments in our revision.*

---

> > ### Comment · Reviewer_xpWW · 2025-08-07
> >
> > I wish to thank the authors for addressing all my points. I have now raised my rating by one point.

---

> > > ### Author Response · Authors · 2025-08-07
> > >
> > > Thank you so much for your thorough review and precious feedback. We will make sure to revise our manuscript by following your insightful suggestions to further improve the quality and clarity of the paper.

---

### Official Review · Reviewer_wLru · 2025-07-03

**Clarity:** 3
**Significance:** 2
**Originality:** 3
**Rating:** 5
**Confidence:** 3

**Summary:**

The authors propose viewing hippocampal place cells not as independent Gaussian bumps but as a population embedding whose inner product reproduces multi-step symmetric random walk transition probabilities. Experiments on open fields, U, S and four-room mazes show high correlation between learned and what the authors consider ideal kernels and high navigation success. The framework generates qualitative phenomena such as scale-dependent field sizes, boundary following and remapping after obstacle removal.

**Questions:**

You initialised the one-step transition matrix using the four‐nearest neighborhood on the lattice. From a neurobiological standpoint, do head-direction modulation or the hexagonal symmetry of grid cells argue for a different neighborhood structure?

Empirically, rodent place fields often shrink, elongate or duplicate near walls and obstacles. Can your representation somehow capture such distortions?

Weber-Fechner law predicts that the tuning width of place cells grows with distance from salient cues or landmarks (at least in the absence of visual inputs that could provide anchoring). Can your hierarchy reproduce a systematic widening of fields with distance?

During inference, you evaluate at continuous coordinates, yet learning was performed on a discrete 40 by 40 lattice. Could you elaborate on how you interpolate or extrapolate between lattice points?

How might the model integrate self-motion cues (path integration) with the learned transition kernel, and what predictions would this make for place-cell dynamics during real-world locomotion (e.g., phase precession)?

**Ethical Concerns:**

["NO or VERY MINOR ethics concerns only"]

**Final Justification:**

I think this is a valuable contribution. The authors have addressed my concerns appropriately and it seems to me that they have done a reasonable job of addressing concerns from other reviewers.

**Limitations:**

Yes

**Paper Formatting Concerns:**

No issues.

**Quality:**

3

**Strengths And Weaknesses:**

Strengths:

Elegant unification of heat-kernel diffusion, spectral embeddings and hippocampal population coding.

Multi-scale hierarchy mirrors dorsoventral field scaling;

Matrix squaring gives $O(log⁡ t)$ scaling.

Path planning uses only local gradients - no global graph search.


Weaknesses:

All tasks are small 2-D lattices with full observability. No comparison with stronger planners (A*, DWA, agents based on successor representations, deep RL) or with SLAM-style real-world data.

The model requires the entire transition matrix to form $q$ before learning; this is unrealistic for animals and for large robotics maps. Can the factorisation be updated online from sampled transitions?

---

> ### Author Rebuttal · Authors · 2025-07-31
>
> Thank you for noting our elegant unification of diffusion, spectral embeddings, and hippocampal coding. We are grateful that you recognize our multi-scale hierarchy and the efficiency of matrix squaring for local gradient-based path planning.
>
> ### Weaknesses
>
> >**1. All tasks are small 2-D lattices with full observability. No comparison with stronger planners (A\*, DWA, agents based on successor representations, deep RL) or with SLAM-style real-world data.**
>
> We thank the reviewer for this important point. The primary focus of this work is to introduce a biologically-motivated theoretical framework and demonstrate that several complex features of hippocampal coding can emerge naturally from the simple principle of learning multi-scale transition probabilities. Our goal was to establish a proof-of-concept for this new perspective rather than to develop a state-of-the-art planner for robotics.
>
> We agree that a comprehensive comparison with established planning algorithms is a crucial next step. To demonstrate this we have conducted a comparison with A* on Success weighted by inverse Path Length (SPL) as suggested.
>
> | Environment | Bug algorithm | A* algorithm |
> |-------------|----------------|-------------------|
> | Open field  | 0.991±0.04    | 0.991±0.04 |
> | U-shape     | 0.919±0.25    | 0.801±0.254 |
> | S-shape     | 1.392±0.98    | 0.915±0.42 |
> | Four-room   | 1.519±1.17    | 0.935±0.029 |
>
>
> The results show that while A* remains the optimal planner, our method achieves highly competitive performance. Our objective $q(y|x, t)$ captures geodesic distance at small t and topological connectivity at large t. For short ranges, our method follows geodesic shortest paths; for long ranges, it prioritizes topological connectivity.
>
> We thank the reviewer again for the suggestion and will add this to our revised manuscript.
>
>
> >**2. The model requires the entire transition matrix to form $q$ before learning; this is unrealistic for animals and for large robotics maps. Can the factorisation be updated online from sampled transitions?**
>
> The reviewer raises a critical point about the biological and practical plausibility of our learning rule. While the current implementation uses batch learning on a pre-computed transition matrix for simplicity, the underlying mathematical framework is fully compatible with online learning. We envision an online implementation where the embeddings, $h$, are updated incrementally based on single-step transitions sampled by the agent as it explores the environment. This would involve a stochastic gradient descent update to minimize the error between the observed transition and the one predicted by the current embeddings. This would be a more biologically plausible and scalable approach. We are actively developing this online version of the model and will revise the discussion to elaborate on this possibility and its implications for online learning and adaptation.
>
> ### Questions
>
> >**1. You initialised the one-step transition matrix using the four‐nearest neighborhood on the lattice. From a neurobiological standpoint, do head-direction modulation or the hexagonal symmetry of grid cells argue for a different neighborhood structure?**
>
> This is a very insightful question! The four-neighbor lattice is indeed a simplification. Our framework is flexible and can accommodate more complex neighborhood structures. For instance, one could easily define the one-step transitions on a hexagonal lattice, which would more closely align with the known symmetry and isotropy of grid cell firing fields. Furthermore, incorporating head-direction information could be achieved by modulating the transition probabilities, making transitions in the agent's current heading direction more likely. We hypothesize that these modifications would lead to representations that even more closely mirror the geometric properties of grid and place cells, and we will discuss this in the future work section as a promising extension.
>
> >**2. Empirically, rodent place fields often shrink, elongate or duplicate near walls and obstacles. Can your representation somehow capture such distortions?**
>
> Thank you for this insightful question. Yes, our model can naturally capture these phenomena. The presence of a boundary or obstacle alters the local transition statistics of the random walk. A state next to a wall has fewer available transitions than a state in the middle of an open field. Since our model learns the embeddings directly from these transition probabilities, the learned representations for states near boundaries will be different from those for states in open areas. This results in distortions are shown in the learned place fields in Figure 2. This is an emergent property, not an explicitly programmed feature, which we believe is a key strength of our model.
>
> >**3. Weber-Fechner law predicts that the tuning width of place cells grows with distance from salient cues or landmarks (at least in the absence of visual inputs that could provide anchoring). Can your hierarchy reproduce a systematic widening of fields with distance?**
>
> This is an insightful question that connects directly to the multi-scale nature of our model. Our hierarchy of representations, learned at different time scales $t$, provides a principled explanation for this phenomenon. The place fields corresponding to longer time scales ($t$) are inherently wider, as they represent transition probabilities over greater distances. If we assume that the brain utilizes representations learned at shorter time scales near salient landmarks (for fine-grained navigation) and longer time scales further away, our model directly predicts a systematic widening of place fields with distance from these anchor points. We will add a discussion of this point to the paper, as it provides a novel theoretical account for this well-documented experimental finding.
>
> >**4. During inference, you evaluate at continuous coordinates, yet learning was performed on a discrete 40 by 40 lattice. Could you elaborate on how you interpolate or extrapolate between lattice points?**
>
> We apologize for not making this clearer in the text. To obtain a continuous representation for planning, we use bi-linear interpolation. After the embeddings, $h$, have been learned for every point on the discrete lattice, the embedding for any continuous coordinates is computed as a weighted average of the embeddings of the four nearest lattice points. The contribution of each lattice point's embedding is weighted by the continuous point's proximity to it. This is a standard method for creating a smooth, continuous function from a discrete grid. We will add a clear description of this interpolation process to the methods section. The ability to interpolate the learned $h$ naturally enables our method to plan smooth paths.
>
> >**5. How might the model integrate self-motion cues (path integration) with the learned transition kernel, and what predictions would this make for place-cell dynamics during real-world locomotion (e.g., phase precession)?**
>
> This is a forward-looking question touching on an exciting frontier. We propose that self-motion cues and path integration can be naturally incorporated through the grid cell system. The modules of grid cells form distance-preserving embeddings with different metrics, and their recurrent connections enable path planning, as demonstrated in recent work on neural path integration and spatial representation learning [1,2,3]. In our appendix, we discuss the interactions between place cells and grid cells, where grid cells preserve distance and serve as Euclidean coordinates, while place cells preserve proximity $q(y|x, t)$ and guide path planning.
>
> Regarding theta phase precession, we have developed an interesting theoretical framework detailed in Appendix F of our submission. Phase precession describes how the timing of place cell spikes advances relative to the theta rhythm as an animal crosses a place field. In our proposed theory, the theta phase of a place cell depends on the inner product between the embedding of the current location and the embedding of the center of the place field of this place cell. Therefore, the theta phase encodes adjacency to the center of the place field. Since place fields of different place cells tile the environment, the theta phases give the agent the awareness of its adjacency to different places.
>
>
> [1] Sorscher, B., Mel, G., Ganguli, S., & Ocko, S. (2019). A unified theory for the origin of grid cells through the lens of pattern formation. Advances in neural information processing systems, 32.
>
> [2] Gao, R., Xie, J., Wei, X. X., Zhu, S. C., & Wu, Y. N. (2021). On path integration of grid cells: group representation and isotropic scaling. Advances in Neural Information Processing Systems, 34, 28623-28635.
>
> [3] Xu, D., Gao, R., Zhang, W., Wei, X. X., & Wu, Y. N.(2024). On Conformal Isometry of Grid Cells: Learning Distance-Preserving Position Embedding. In The Thirteenth International Conference on Learning Representations.
>
> *Thanks again for your thoughtful evaluation and constructive suggestions. We will revise our paper by incorporating all your insightful comments and suggestions to improve the clarity and rigor of our work.*

---

> > ### Comment · Reviewer_wLru · 2025-08-06
> >
> > Thank you for the detailed response.
> >
> > To contextualize results from A*, it might be useful to show values for random or some other type of exploration (for example, for a four-room environment, the difference between the two algorithms is 0.6 and while I agree that it's expected for A* to perform better, it would be informative to have more baselines to better interpret the magnitude of the difference). I'm confident this is something the authors can easily do for the final version of the manuscript.
> >
> > Regarding question 2, it would be informative if you could demonstrate the approach on realistic data, e.g., some of the classical work from O'Keefe & Neil Burgess (Fig. 2 in https://www.nature.com/articles/381425a0)

---

> ### Author Response · Authors · 2025-08-07
>
> > **To contextualize results from A\*, it might be useful to show values for random or some other type of exploration (for example, for a four-room environment, the difference between the two algorithms is 0.6 and while I agree that it's expected for A\* to perform better, it would be informative to have more baselines to better interpret the magnitude of the difference). I'm confident this is something the authors can easily do for the final version of the manuscript.**
>
> Thank you for the valuable advice! Following your advice, we have compared our method with several baselines in the Four-Room environment, with each trial limited to 50,000 steps. The results show that baseline approaches like the Bug algorithm (without oracle guidance) and Random Walk fail to solve the most of the challenging scenarios.
>
> | Metric | Bug (w/ Oracle) | A* Search | Bug (w/o Oracle) | Random Walk | **Our Method** |
> | :------------------- | :-------------: | :---------: | :----------------: | :------------: | :--------------: |
> | **Success Rate (%)** | 100 | 100 | 14 | 2 | 100 |
> | **SPL (↓)** | 1.52 ± 1.17 | 0.94 ± 0.03 | 0.93 ± 0.43 | 95.26 ± 0.00 | 1.00 |
>
> > **Regarding question 2, it would be informative if you could demonstrate the approach on realistic data, e.g., some of the classical work from O'Keefe & Neil Burgess (Fig. 2 in https://www.nature.com/articles/381425a0)**
>
> Thank you for directing us to this classical paper! Following the experimental design in [4], we have implemented two key environments: a rectangular environment (40×80 grids, analogous to their HR condition) and a larger square environment (80×80 grids, analogous to their LS condition).
>
> Using t=128 iterations, our method successfully reproduces key phenomena reported in their Figures 1 and 2: (1) field elongation in the rectangular environment along the extended dimension, and (2) field dissection into two components in the larger square environment.
>
> The emergence of elongated patterns in HR and dissected fields in LS demonstrates that our method can model the geometric determinants of place field structure described in this seminal work. These results show clear correspondence with their experimental observations of how environmental geometry shapes hippocampal spatial representations. Due to rebuttal policy constraints, we are not allowed to present addtional plots, but we will include these comparative analyses and visualizations in the final version of the manuscript.
>
> [4] O'Keefe, J., & Burgess, N. (1996). Geometric determinants of the place fields of hippocampal neurons. Nature, 381(6581), 425-428.
>
> *Thank you so much for your insightful guidance!*

---

### Official Review · Reviewer_qzbz · 2025-07-03

**Clarity:** 3
**Significance:** 3
**Originality:** 3
**Rating:** 5
**Confidence:** 4

**Summary:**

This paper proposes that at a population level, the place system encodes location and transition probabilities for local motion. Population level responses are optimized such that the the similarity between responses at positions $x$ and $y$ (given by the inner product of the population level response at these locations) matches the transition probability between those locations $p(y(t)|x(0))$ for a given timescale $t$, where this transition probability is assumed to match that for an unbiased random walk on the environment. Under very mild assumptions, optimizing the population level responses in this way leads to place cell representations. Optimizing jointly over multiple time scales leads to multiple place cell scales. These representations can be used for efficient path planning.

**Questions:**

Most of my comments / questions are relatively minor:
* The meaning in the abstract is a little bit unclear before reading the paper, it might be good to use fewer equations in the abstract
* How robust is this setup to noisy place cell responses? I.e., if the embedding $h$ is noisy due to firing rate variability, what changes, and how much noise can one get away with?
* The setup was a little bit confusing on first reading. $t$ is meant to represent a time scale for which transition kernels are optimized for, but at first it may be mistakenly be thought that the transitions that are optimized for are changing (and growing) with time for an agent, which might lead to confusion. Changing text slightly to emphasise what is meant by time $t$ or changing the notation slightly  ( $t\rightarrow \tau$ or $p(y|x,t)\rightarrow p(y(t+t')|x(t'))$ for example) might clarify the meaning for readers in the early sections of the paper.
* There is experimental evidence of place cells that respond to location and travel direction (i.e., they respond when an agent travels through a particular location from North to South but not from others). Would you expect to see place cells with this property if assumptions on the transition kernel being symmetric are lifted, and the underlying random walk is biased in some way?
* The representations are optimized for a random walks on space, where $p(y(t)|x(0))$ is gaussian with $\sigma \sim \sqrt{t}$. However, many exploration strategies are not truely random walks, but biased in some way. Would one expect this scaling relation to change?

**Ethical Concerns:**

["NO or VERY MINOR ethics concerns only"]

**Final Justification:**

I apologize for the late response to the authors rebuttal. I had only a few very minor gripes with the clarity of the description, all of which were addressed in the rebuttal, so I will maintain my high initial score, as I believe this paper will be of value to the neurips community.

**Limitations:**

See above

**Quality:**

4

**Strengths And Weaknesses:**

Strengths:
* There is a lot of experimental evidence that hippocampal coding is predictive, so theoretical work that frames hippocampal coding explicitly in terms of maximizing transition probabilities is well motivated and also understudied
* With mild assumptions, many relevant features of hippocampal coding are naturally recovered.
* Figures are well made and clear

Weaknesses:
* It would be nice if they authors discussed connections to known experimental results in a little bit more depth
* The model is obscured a little bit by notation

---

> ### Author Rebuttal · Authors · 2025-07-31
>
> We appreciate your recognition of our theoretical framework's strong motivation, its recovery of key hippocampal features, and the clarity of our figures.
>
> ### Weaknesses
>
> > **1. It would be nice if they authors discussed connections to known experimental results in a little bit more depth**
>
> We appreciate this suggestion. In the revised manuscript, we will expand our discussion of connections to experimental results. For instance, in our response to Reviewer wLru's third question, we elaborate on how our multi-scale hierarchy naturally reproduces the systematic widening of place fields with distance from salient cues, consistent with the Weber-Fechner law. We also discuss in our response to Reviewer wLru's second question how our model spontaneously captures the empirically observed distortions (shrinking, elongating) of place fields near boundaries and obstacles. Furthermore, the adaptive scale selection mechanism inherently mirrors the dorsoventral axis scaling observed in the hippocampus. These points will be highlighted in the relevant sections to more explicitly link our theoretical findings to neurobiological evidences.
>
>
> >**2. The model is obscured a little bit by notation**
>
> We agree with this and will make sure to clarify the notation and explanations throughout the paper, especially in the early sections. As suggested by your third question, we have changed the notation for the time scale from $t$ to $\tau$ to avoid confusion with the agent's real-time movement. This change, along with revised explanations, aims to make the model's concepts more accessible.
>
>
> ### Questions
>
>
> >**1. The meaning in the abstract is a little bit unclear before reading the paper, it might be good to use fewer equations in the abstract**
>
> We thank the reviewer for this suggestion. In the revised manuscript, we will reduce the number of equations in the abstract and focus on a more descriptive and intuitive explanation of our core ideas. We will emphasize the conceptual framework of our model – viewing the hippocampus as a predictive engine that learns multi-scale representations of space – and how this leads to the emergent properties we observe.
>
>
> >**2. How robust is this setup to noisy place cell responses? I.e., if the embedding is noisy due to firing rate variability, what changes, and how much noise can one get away with?**
>
> This is an excellent question. To address this, we performed additional experiments by adding Gaussian noise to the learned embeddings, $h$, during the path-planning phase. And we test the success rate of path planning under multiple noise levels in different environments. Since $\|h\|^2=1$ due to normalization, to be more systematic, we measure the noise levels with $\sigma^2*n$ in the table, where $n$ is the number of cells. Our preliminary results shown in the table below indicate that the model is robust to moderate levels of noise.
>
> |            | Open field | S-shape |
> |-----------------|------|----------|
> |  No noise       | 100% | 100% |
> |  $\sigma^2*n=0.05$  | 98%  | 96%  |
> |  $\sigma^2*n=0.2$  | 90%  | 90%  |
>
>
>
> >**3.The setup was a little bit confusing on first reading. t is meant to represent a time scale for which transition kernels are optimized for, but at first it may be mistakenly be thought that the transitions that are optimized for are changing (and growing) with time for an agent, which might lead to confusion. Changing text slightly to emphasise what is meant by time t or changing the notation slightly (t->$\tau$ or p(y|x,t)->p(y(t+t')|x(t')) for example) might clarify the meaning for readers in the early sections of the paper.**
>
>
> We are grateful to the reviewer for pointing out this potential source of confusion. To improve clarity, we will take the reviewer's suggestion and change the notation for the time scale from $t$ to $\tau$ throughout the paper. This will help to distinguish the time scale of the transition kernels from the time variable of the agent's movement. We will also revise the text in the early sections of the paper to explicitly state that $\tau$ represents a fixed time scale for which the representations are learned.
>
> >**4. There is experimental evidence of place cells that respond to location and travel direction (i.e., they respond when an agent travels through a particular location from North to South but not from others). Would you expect to see place cells with this property if **assumptions on the transition kernel being symmetric are lifted**, and the underlying random walk is biased in some way?**
>
> This is a fascinating question that points to an exciting future direction for our work. The reviewer is correct that our current model, with its symmetric transition kernel, would not produce direction-tuned place cells. However, we strongly agree that lifting the assumption of a symmetric transition kernel is a promising way to model such cells.
>
> If the underlying random walk were biased (e.g., due to a prevailing wind, a slope in the environment, or an animal's natural tendency to move in a particular direction), the transition probabilities would become asymmetric. In this case, we could extend our framework to learn $p(y|x, t) = ⟨h(x, t), w(y, t)⟩$ to accommodate asymmetry, where $w(y, t)$ can be interpreted as connection weights that differ from the source embeddings $h(x, t)$. This formulation would naturally capture directional biases while maintaining the inner product structure that makes our model computationally tractable.
>
> With such asymmetric transition probabilities, our model would learn embeddings that reflect this asymmetry, and we would expect to see the emergence of place cells that are tuned to both location and direction of travel. We will add a discussion of this point to the future work section of the paper and acknowledge this as a key extension of our framework.
>
> >**5. The representations are optimized for a random walks on space, where $p(y(t)|x(0))$ is gaussian with $\sigma \sim \sqrt{t}$. However, many exploration strategies are not truely random walks, but biased in some way. Would one expect this scaling relation to change?**
>
> The reviewer raises a very important point. While the random walk is a useful starting point, we agree that real-world exploration is often biased. We do expect the scaling relation to change with different exploration strategies. For example, a Levy flight, which involves occasional long-distance jumps, would lead to a different scaling of the transition probabilities.
>
> Our framework is flexible enough to accommodate different exploration strategies. The key is that as long as the exploration strategy can be described by a transition kernel, our model can learn the corresponding representations. We believe that investigating how different exploration strategies affect the learned representations is a rich area for future research. We will add a paragraph to the discussion section to address this point and speculate on how the scaling relation might change under different, more realistic exploration models.
>
> *Thank you again for your comprehensive review. We will revise our paper by incorporating all your insightful comments and suggestions to strengthen our paper.*

---

> > ### Comment · Reviewer_qzbz · 2025-08-08
> >
> > Thank you for the detailed response to my questions and comments, and I apologize for the late response. These address all of my (minor) points, and I will maintain my high evaluation.

---

> > > ### Author Response · Authors · 2025-08-08
> > >
> > > We are deeply grateful for your positive review! We will revise our paper based on your insightful comments and questions.

---

### Decision · Program_Chairs · 2025-09-17

**Decision:**

Accept (poster)

**Comment:**

The paper considers a new model of hippocampal place cells using population embeddings with random walk kernels. They demonstrate the suitability of their method in a path modeling framework across extensive simulations, including remapping, to also showcase biological correspondence with real place cells. Reviewers suggested additional connection to known experimental results, relevant methods like SR, and comparisons to other planners; these were clarified and expanded upon by the authors during rebuttal. Reviewers unanimously recommended acceptance.